# Brain endothelial cell TRPA1 channels initiate neurovascular coupling

Pratish Thakore[1], Michael G Alvarado[1], Sher Ali[1], Amreen Mughal[2], Paulo W Pires[3], Evan Yamasaki[1], Harry AT Pritchard[1,4], Brant E Isakson[5,6], Cam Ha T Tran[7], Scott Earley[1]*

[1]Department of Pharmacology, Center for Molecular and Cellular Signaling in the Cardiovascular System, University of Nevada, Reno School of Medicine, Reno, United States; [2]Department of Pharmacology, College of Medicine, University of Vermont, Burlington, United States; [3]Department of Physiology, College of Medicine, University of Arizona, Tucson, United States; [4]Institute of Cardiovascular Sciences, University of Manchester, Manchester, United Kingdom; [5]Department of Molecular Physiology and Biological Physics, University of Virginia, Charlottesville, United States; [6]Robert M. Berne Cardiovascular Research Center, University of Virginia, Charlottesville, United States; [7]Department of Physiology & Cell Biology, Center for Molecular and Cellular Signaling in the Cardiovascular System, University of Nevada, Reno School of Medicine, Reno, United States

*For correspondence:
searley@med.unr.edu

Competing interests: The authors declare that no competing interests exist.

**Abstract** Cerebral blood flow is dynamically regulated by neurovascular coupling to meet the dynamic metabolic demands of the brain. We hypothesized that TRPA1 channels in capillary endothelial cells are stimulated by neuronal activity and instigate a propagating retrograde signal that dilates upstream parenchymal arterioles to initiate functional hyperemia. We find that activation of TRPA1 in capillary beds and post-arteriole transitional segments with mural cell coverage initiates retrograde signals that dilate upstream arterioles. These signals exhibit a unique mode of biphasic propagation. Slow, short-range intercellular $Ca^{2+}$ signals in the capillary network are converted to rapid electrical signals in transitional segments that propagate to and dilate upstream arterioles. We further demonstrate that TRPA1 is necessary for functional hyperemia and neurovascular coupling within the somatosensory cortex of mice in vivo. These data establish endothelial cell TRPA1 channels as neuronal activity sensors that initiate microvascular vasodilatory responses to redirect blood to regions of metabolic demand.

## Introduction

Neurons within the brain lack significant intrinsic energy reserves; thus, the metabolic substrates for energy generation must be delivered to active regions in real time. However, substantial increases in global blood flow to the brain are limited by constraints imposed by the enclosing skull and the finite amount of total blood available for regional distribution. The brain therefore relies on dynamic local distribution of the available blood supply to the sites of highest metabolic activity. This vital process is accomplished by a complex interconnected network of cerebral pial arteries at the surface, parenchymal arterioles that provide a conduit to the interior, and a vast capillary network. Communication between active neurons and the cerebral microvasculature regulates activity-dependent increases in blood perfusion (functional hyperemia) in the brain through processes collectively referred to as 'neurovascular coupling' (NVC) (*Iadecola, 2017*). One conceptual model of NVC that has received considerable research attention postulates that glutamate released during neurotransmission binds to $G_{q/11}$ protein-linked metabotropic glutamate receptors (mGluRs) on perisynaptic astrocytic

processes to stimulate $Ca^{2+}$-dependent signaling pathways that ultimately cause the release of vaso-dilator substances from astrocytic endfeet encircling parenchymal arterioles (*Attwell et al., 2010*; *Filosa et al., 2004*; *Filosa et al., 2006*; *Iadecola and Nedergaard, 2007*). These factors are purported to act directly on smooth muscle cells in the walls of arterioles to cause vasodilation and locally increase blood flow. Several recent studies have challenged this model, including one reporting that mGluR expression in astrocytes is undetectable after postnatal week 3, implying that this mechanism does not function in adults (*Sun et al., 2013*). Other reports have suggested that NVC can occur in the absence of astrocytic $Ca^{2+}$ signaling (*Jego et al., 2014*; *Nizar et al., 2013*) and that astrocytic $Ca^{2+}$ signals lag vasodilator responses recorded in vivo in awake, behaving mice (*Tran et al., 2018*). Longden et al. put forward a new paradigm for NVC, demonstrating that activation of inward-rectifying $K^+$ ($K_{ir}$) channels on brain capillary endothelial cells by extracellular $K^+$ ions, released during neuronal activity, generates propagating electrical signals that cause dilation of upstream parenchymal arterioles (*Longden et al., 2017*). Given the essential nature of the NVC response for healthy brain function and life itself, it is reasonable to presume that multiple sensory modalities with overlapping and complementary roles operate within brain capillary endothelial cells to ensure the fidelity of neuronally driven functional hyperemia.

Here, we tested the hypothesis that transient receptor potential ankyrin 1 (TRPA1) channels in brain capillary endothelial cells contribute to functional hyperemia in the brain. Members of the TRP superfamily of cation channels act as molecular sensors of diverse physical and chemical stimuli (*Flockerzi and Nilius, 2014*). In cerebral pial arteries and parenchymal arterioles, activation of TRPA1 channels by endogenously produced lipid peroxide products, including 4-hydroxynonenal (4-HNE), leads to endothelium-dependent vasodilation (*Earley et al., 2009*; *Pires and Earley, 2018*; *Qian et al., 2013*; *Sullivan et al., 2015*; *Trevisani et al., 2007*). Using a combination of ex vivo and in vivo approaches, we find that activation of endothelial cell TRPA1 channels within capillaries, including post-arteriole transitional segments covered by enwrapping mural cells, generates retrograde propagating signals that lead to the dilation of upstream parenchymal arterioles. Specifically, we demonstrate that $Ca^{2+}$ signals initiated by TRPA1 channels slowly propagate through the capillary network by a previously unknown pathway that is dependent on ionotropic purinergic $P_2X$ receptor activity and pannexin-1 (Panx1) expression. Within the post-arteriole region, slowly propagating $Ca^{2+}$ signals are converted to rapidly propagating electrical signals that cause dilation of upstream arterioles. Our data also show that endothelial cell TRPA1 channels contribute to blood flow regulation and functional hyperemia in the somatosensory cortex in vivo. We conclude that, under physiological conditions, TRPA1 channels in the brain endothelium are capable of mediating NVC.

## Results

### Functional TRPA1 channels are present in native brain capillary endothelial cells

Native cerebral capillary endothelial cells were freshly isolated using combined mechanical and enzymatic dispersion (*Longden et al., 2017*) and were patch-clamped in the conventional whole-cell configuration. Increasing extracellular $[K^+]$ to 60 mM stimulated $Ba^{2+}$-sensitive $K_{ir}$ currents in these cells, whereas NS309, a selective activator of small- and intermediate-conductance $Ca^{2+}$-activated $K^+$ channels (SK and IK, respectively) had no effect (*Figure 1—figure supplement 1A to D*). These attributes are consistent with those reported for native brain capillary endothelial cells in a prior study (*Longden et al., 2017*).

Non-selective cation currents in native brain capillary endothelial cells were recorded by patch-clamp electrophysiology in the whole-cell configuration using symmetrical cation gradients established by intracellular and extracellular solutions. In isolated cells, application of voltage ramps from −80 to +80 mV over 300 ms in the presence of the endogenous TRPA1 agonist 4-HNE (100 nM) elicited dually rectifying currents with a reversal potential near 0 mV (*Figure 1A and B*). The selective TRPA1 inhibitor HC-030031 (10 μM) significantly attenuated this current. To confirm that 4-HNE actions are specific for TRPA1, we generated endothelial cell-specific TRPA1-knockout (*Trpa1*-ecKO) mice. This was accomplished by initially crossing mice homozygous for *loxP* sites flanking the region within *Trpa1* encoding the S5 and S6 transmembrane domains (*Trpa1$^{fl/fl}$*) with mice heterozygous for

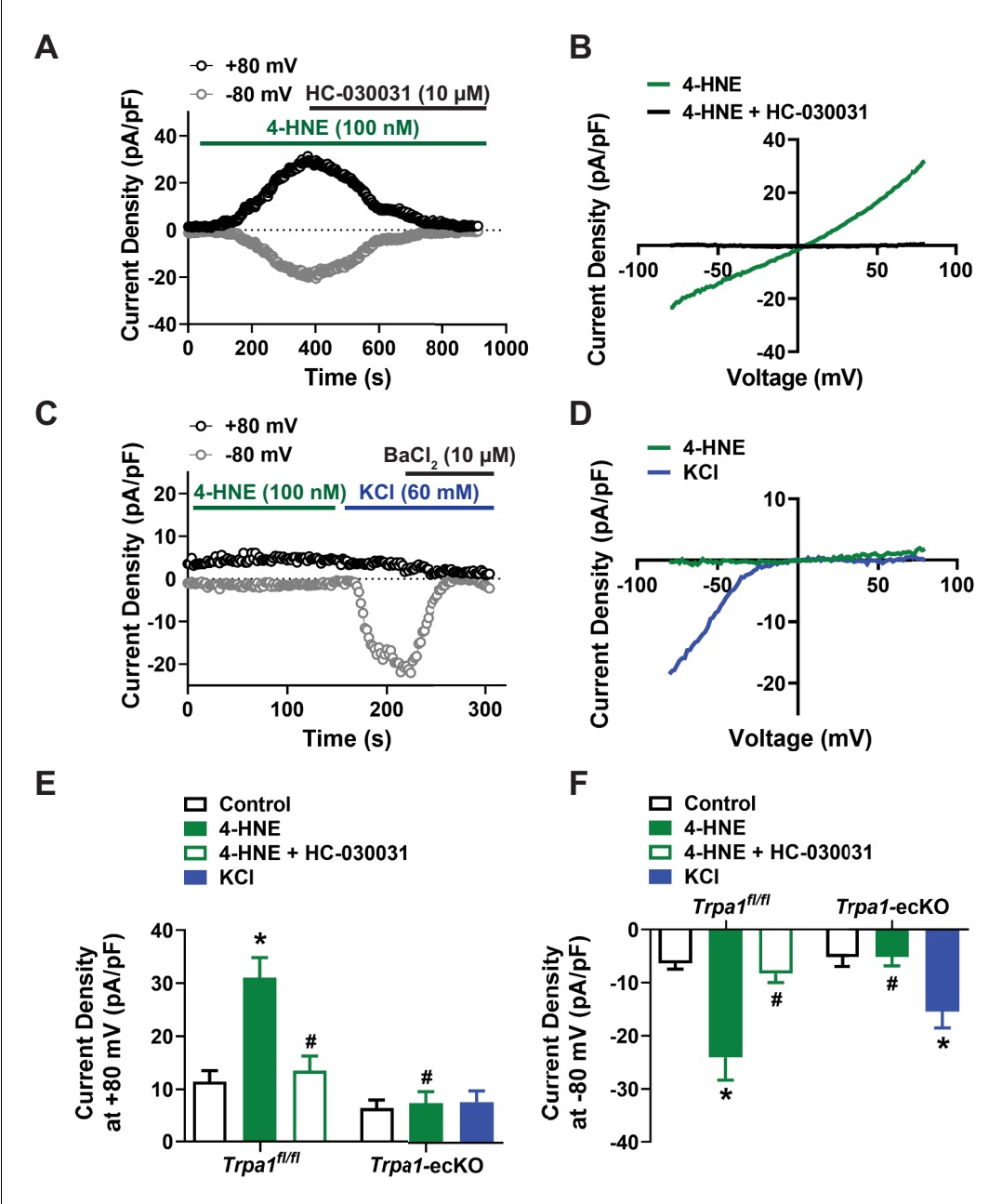

**Figure 1.** TRPA1 channels are functionally expressed in native brain capillary endothelial cells. (A and B) Representative current versus time trace (A) and *I-V* relationship (B) from a whole-cell patch-clamp electrophysiology experiment demonstrating that the TRPA1 activator 4-HNE (100 nM) elicited a current in a native capillary endothelial cell isolated from a control *Trpa1*$^{fl/fl}$ mouse that was blocked by the selective TRPA1 antagonist HC-030031 (10 μM). (C and D) Representative current versus time trace (C) and *I-V* relationship (D) demonstrating that 4-HNE was unable to elicit a current in a native capillary endothelial cell from a *Trpa1*-ecKO mouse. Cell viability was confirmed by evoking K$_{ir}$ currents with raised extracellular KCl (60 mM) that were sensitive to BaCl$_2$ (10 μM). (E and F) Summary data showing that the TRPA1 current produced by 4-HNE at +80 mV (E) and −80 mV (F) was blocked by HC-030031 and was absent in cells from *Trpa1*-ecKO mice (*Trpa1*$^{fl/fl}$, n = 12 cells from five animals; *Trpa1*-ecKO, n = 8 cells from three animals; *p<0.05, one-way ANOVA).

The online version of this article includes the following source data and figure supplement(s) for figure 1:

**Source data 1.** Individual data points and analysis summaries for datasets shown in *Figure 1*.

**Figure supplement 1.** Lack of functional IK and SK channels in native capillary endothelial cells.

**Figure supplement 1—source data 1.** Individual data points and analysis summaries for datasets shown in *Figure 1—figure supplement 1*.

TEK tyrosine kinase (*Tek*) promoter-driven expression of *Cre*-recombinase (*Tek$^{Cre}$*). Intermediate heterozygote mice were crossed to generate *Trpa1*-ecKO mice. Our prior study demonstrated that TRPA1 expression is undetectable by western blotting in cerebral arteries from *Trpa1*-ecKO mice, but expression was unaffected in other cell types (*Sullivan et al., 2015*). *Trpa1$^{fl/fl}$* mice, homozygous for floxed *Trpa1* but without expression of *Cre*-recombinase, were used as controls. We found that 4-HNE did not stimulate cation currents in brain capillary endothelial cells isolated from *Trpa1*-ecKO mice (*Figure 1C to F*), supporting the selectivity of 4-HNE and the expression of functional TRPA1 channels in native brain capillary endothelial cells from wild-type and *Trpa1$^{fl/fl}$* mice. Control patch-clamp experiments in which extracellular [K$^+$] was raised to 60 mM to evoke K$_{ir}$ currents (*Longden et al., 2017*) further confirmed the viability of capillary endothelial cells isolated from *Trpa1*-ecKO mice (*Figure 1C to F*).

## Stimulation of TRPA1 channels in brain capillary endothelial cells induces dilation of upstream parenchymal arterioles

To determine if activation of capillary endothelial cell TRPA1 channels relaxes the upstream vasculature, we utilized a recently described ex vivo cerebral microvascular preparation in which microvascular segments with intact capillary branches are isolated from the brain, cannulated with micropipettes, and pressurized to 40 mmHg (*Longden et al., 2017*; *Figure 2A*). Preparations varied in terms of length of the post-arteriole transitional segment and number of capillary branches present with preparations predominantly exhibiting primary and secondary branches that occasionally extended beyond the tertiary branch order. In these experiments, a micropipette attached to a picospritzer was placed adjacent to the capillary extremities, which were stimulated by locally applying pulses (7 s; pressure,~10 psi) of various substances dissolved in artificial cerebral spinal fluid (aCSF). The spread of the picospritzed solution was assessed by pulsing aCSF containing 1% (w/v) Evans Blue dye. Control experiments showed that dye ejected from the micropipette was localized to the capillary region and did not spread to the upstream parenchymal arteriole segment (*Video 1*). To validate the viability of this preparation, we focally applied pulses of aCSF containing elevated [K$^+$] (10 mM) onto capillary extremities. This maneuver induced a transient, reversible, and reproducible dilation of the upstream arteriole segment, as has been shown previously (*Longden et al., 2017*). This response was attenuated by incubating the ex vivo preparation with BaCl$_2$ (30 µM) to inhibit K$_{ir}$ channels (*Figure 2—figure supplement 1A and B*). Picospritzing capillary extremities with standard aCSF (3 mM K$^+$) did not affect the diameter of upstream arteriole segments (*Figure 2—figure supplement 1C and D*). Additional control studies comparing the dilatory response of upstream arterioles to elevated [K$^+$] application before and after severing capillaries from the upstream arteriole segment (*Figure 2—figure supplement 2A*) showed that interruption of this connection prevented elevated [K$^+$]-induced dilation of upstream arterioles (*Figure 2—figure supplement 2B*). These data demonstrate the viability of our microvascular preparation and validate conducted vasodilation in response to stimulation of capillaries with elevated [K$^+$], as previously reported by *Longden et al., 2017*.

TRPA1 channels in capillary segments of the ex vivo microvascular preparation were stimulated by focally applying pulses of various TRPA1-activating compounds dissolved in aCSF. Application of 4-HNE onto capillary extremities dilated upstream arterioles in ex vivo preparations from wild-type mice (*Figure 2B*) in a concentration-dependent manner. The EC$_{50}$ for 4-HNE-induced dilation was 190 nM, and the maximal response was achieved at 1 µM; thus, this latter concentration was used in subsequent investigations (*Figure 2C*). Dilation of parenchymal arterioles in response to capillary-applied 4-HNE was significantly attenuated by superfusing the ex vivo preparation with HC-030031 (*Figure 2B and D*). Application of the TRPA1-activating compound, allyl isothiocyanate (AITC), to capillary beds also produced a concentration-dependent dilation of upstream arterioles (*Video 2*). The EC$_{50}$ of AITC-induced arteriole dilation was 6 µM, and the maximal response was achieved at a concentration of 30 µM (*Figure 2E*). HC-030031 also inhibited upstream dilation in response to AITC (30 µM) (*Figure 2F*). In addition, the dilatory response of upstream arterioles to stimulation of capillary extremities with AITC was absent after severing the connection between capillaries and the upstream arteriole (*Figure 2—figure supplement 2C*), providing evidence that this response requires intercellular signal propagation. In ex vivo preparations obtained from *Trpa1*-ecKO mice, stimulation of capillaries with elevated [K$^+$] evoked dilation of upstream arterioles (*Figure 2G to I*); however, neither 4-HNE (*Figure 2G and H*) nor AITC (*Figure 2I*) had any effect at concentrations

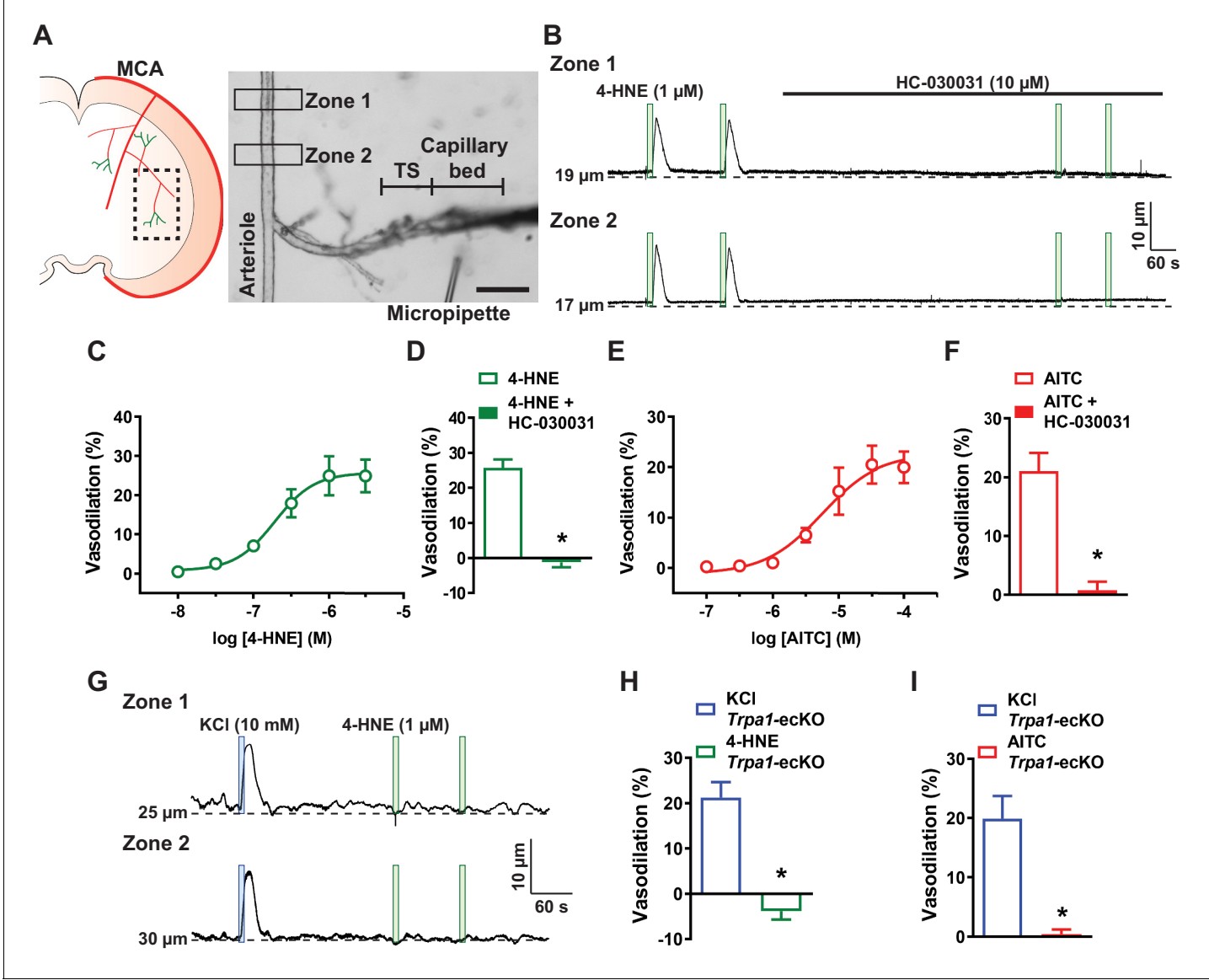

**Figure 2.** Capillary TRPA1 channels initiate conducted vasodilation in the cerebral microcirculation. (A) Experimental illustration. Microvascular preparations were obtained from the subcortical region that is supplied by the middle cerebral artery (MCA). Representative image of an ex vivo microvascular preparation consisting of an intact parenchymal arteriole with attached capillaries. The post-arteriole transitional segment (TS) and capillaries are located at the distal end of the offshoot arteriole branch. Scale bar = 100 µm. Drugs were directly applied onto capillary extremities with a micropipette. (B) Representative traces showing that application of 4-HNE (1 µM; green box) directly onto capillary extremities produced a reproducible increase in lumen diameter in two different areas (Zone 1 and Zone 2) on the upstream arteriole segment in microvascular preparations isolated from wild-type mice. The dilation produced by 4-HNE was blocked the selective TRPA1 antagonist HC-030031 (10 µM). (C) Concentration-response curve produced by locally applying 4-HNE to capillary extremities over a concentration range of 10 nM to 30 µM (n = 6 preparations from five animals). An EC$_{50}$ of 190 nM was calculated from the plot of a non-linear regression curve. (D) Summary data showing that dilation to 4-HNE (1 µM) was blocked by HC-030031 (10 µM) (n = 9 preparations from six animals; *p<0.05, paired *t*-test). (E) Concentration-response curve produced by locally applying the TRPA1 agonist AITC to capillary extremities over a concentration range of 100 nM to 100 µM. An EC$_{50}$ of 6 µM was calculated from the plot of a non-linear regression curve (n = 6 preparations from five animals). (F) Summary data showing that dilation to AITC (30 µM) was blocked by HC-030031 (10 µM) (n = 7 preparations from five animals; *p<0.05, paired *t*-test). (G) Representative traces showing that application 4-HNE (1 µM; green box) onto capillary extremities was unable to dilate the upstream arteriole in microvascular preparations from *Trpa1*-ecKO mice, whereas application of KCl (10 mM; blue box) effectively dilated the upstream arteriole in these preparations. (H and I) Summary data showing that neither 4-HNE (1 µM) (n = 7 preparations from five animals; *p<0.05) (H) nor AITC (30 µM) (n = 7 preparations from four animals; *p<0.05, paired *t*-test) (I) evoked dilation of upstream arterioles in preparations from *Trpa1*-ecKO mice.

The online version of this article includes the following source data and figure supplement(s) for figure 2:

**Source data 1.** Individual data points and analysis summaries for datasets shown in *Figure 2*.

*Figure 2 continued on next page*

*Figure 2 continued*

**Figure supplement 1.** KCl-induced dilation of upstream arterioles is blocked by BaCl₂.
**Figure supplement 1—source data 1.** Individual data points and analysis summaries for datasets shown in *Figure 2—figure supplement 1*.
**Figure supplement 2.** Severing the connection between capillaries and the arteriole segment of an ex vivo microvascular preparation.
**Figure supplement 2—source data 1.** Individual data points and analysis summaries for datasets shown in *Figure 2—figure supplement 2*.

that produced maximal dilation of microvascular preparations from control animals. Together, these data demonstrate that activation of TRPA1 channels in brain capillary endothelial cells produces a signal that propagates to upstream parenchymal arterioles to cause dilation.

## Biphasic propagation velocity of conducted vasodilator signals initiated by TRPA1

We next used a fluorescence labeling strategy to image the cell types present in our ex vivo cerebral microvascular preparation. Cannulated preparations were stained with isolectin B4 to visualize the endothelium and anti-α-smooth muscle actin antibody to visualize mural cell coverage. Fluorescent images obtained showed an interconnected network of endothelial cells encompassed by mural cells (*Figure 3A* to C). In agreement with prior studies (*Grant et al., 2019*; *Hartmann et al., 2015*; *Hill et al., 2015*), distinct variations in mural cell morphology were observed in different vascular segments. Endothelial cells of the parenchymal arteriole are encased by a continuous circumferential layer of vascular smooth muscle cells, whereas post-arteriole transitional segments (i.e. segments between parenchymal arterioles and the dense capillary region) are covered by 'ensheathing pericytes' that are morphologically distinct from vascular smooth muscle cells and identified by the classic bump-on-a-log characteristic (*Grant et al., 2019*; *Hartmann et al., 2015*; *Hill et al., 2015*). Mural cell coverage of endothelial cells of the capillary bed was incomplete, such that only a few capillary pericytes or thin-stranded capillary processes, if any, were observed in our isolated ex vivo preparations (*Grant et al., 2019*; *Hartmann et al., 2015*; *Hill et al., 2015*). To visualize the elastin layer present only in arteries and arterioles, preparations were stained with Alexa Fluor 633 hydrazide which specifically binds to elastin fibers (*Shen et al., 2012*). Elastin fibers were observed in the arteriole segment of our microvascular preparations, but not in the transitional segment or the capillary region (*Figure 3D*). We next compared the propagation velocity of TRPA1-induced conducted vasodilator signals in vascular segments completely covered by mural cells (arterioles and post-arteriole transitional segments) with that in sparsely covered capillaries. The propagation velocity of the vaso-

dilator signal is defined here as the time interval between stimulation of capillaries and start of dilation of the upstream arteriole, normalized to the distance between the site of stimulation and the point where arteriole dilation was measured. We found that the propagation velocity of signals initiated by stimulation of TRPA1 channels in capillaries was significantly slower than that elicited by stimulation of K_{ir} channels (*Figure 3E*). These data imply that TRPA1 channels in brain capillary endothelial cells mediate a signal that dilates upstream arterioles through a mechanism distinct from that initiated by K_{ir} channels. An analysis of TRPA1-dependent responses in different segments of the cerebral microvasculature showed that the propagation velocity of vasodilator signals from the post-arteriole transitional region (first vascular segment with mural cell coverage) to the upstream parenchymal arteriole was comparable to that of signals produced in this segment by picospritzing capillaries with elevated [K⁺] (*Figure 3F*).

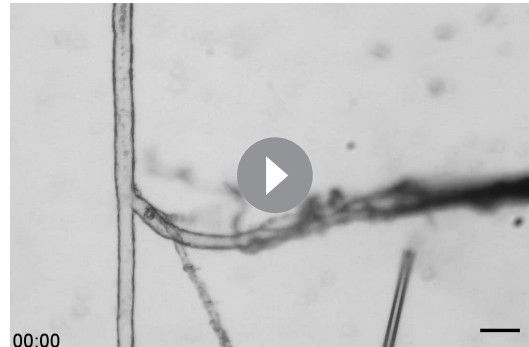

**Video 1.** Localized application of Evans Blue dye onto capillary extremities of a microvascular preparation. Representative time-series images of a microvascular preparation demonstrating that application of Evans Blue dye (1% w/v) is localized to the region of the capillary tree and does not spread to the upstream parenchymal arteriole. Evans Blue was applied after 10 s. Scale bar = 100 µm.
https://elifesciences.org/articles/63040#video1

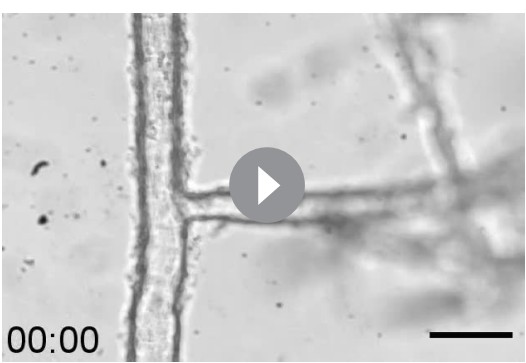

**Video 2.** Localized application of AITC onto capillary extremities of a microvascular preparation dilates the upstream arteriole. Representative time-series images of a microvascular preparation demonstrating that localized application of AITC (30 µM) onto capillary extremities dilates the upstream arteriole. AITC was applied after 10 s. Scale bar = 50 µm.
https://elifesciences.org/articles/63040#video2

Consequently, we infer that the signal produced by activation of TRPA1 channels propagates through the capillary network from the site of initiation to the transitional region more slowly than the fast-electrical signal initiated by activation of $K_{ir}$ channels. We propose a biphasic propagation model of conducted vasodilator signaling in which activation of TRPA1 channels in capillary endothelial cells generates a slowly propagating intercellular signal that is converted to a rapidly conducted electrical signal in vascular segments fully covered by mural cells (*Figure 3G*).

## Slow-phase propagation of vasodilator signals through the capillary network requires $Ca^{2+}$ signals generated by endothelial cell Panx1 channels and ATP

Our next goal was to identify the mechanisms underlying slow intercellular propagation of vasodilator signals initiated by TRPA1 channels within regions of the capillary network more distal to the feeding arteriole. Pannexin proteins are centrally important for intercellular signaling (*Barbe et al., 2006*). Accordingly, we probed the role of pannexins in conducted vasodilator responses using cerebral microvascular preparations from tamoxifen-inducible, endothelial cell-specific *Panx1*-knockout (*Panx1*-ecKO) mice (*Lohman et al., 2015*). This was accomplished by initially crossing mice homozygous for *loxP* sites flanking exon 3 of *Panx1* (*Panx1*$^{fl/fl}$) with mice heterozygous for inducible vascular endothelial cadherin (*Cdh5*)-*Cre*. Intermediate heterozygote mice were crossed to generate *Panx1*-ecKO mice. We found that dilation of upstream parenchymal arterioles following focal application of 4-HNE or AITC to capillary extremities was significantly blunted in ex vivo preparations obtained from *Panx1*-ecKO mice compared with those from tamoxifen-injected control mice expressing only *Cdh5-Cre* as well as vehicle (peanut oil)-injected *Panx1*-ecKO mice (*Figure 4A* to C). In contrast, conducted vasodilation initiated by stimulation of $K_{ir}$ channels in brain capillary endothelial cells was not affected by endothelial cell *Panx1* knockout (*Figure 4D*). These data demonstrate that TRPA1 channels in brain capillary endothelial cells initiate conducted vasodilator responses through an endothelial cell Panx1-dependent mechanism that is independent of the response initiated by $K_{ir}$ channels.

Prior studies have demonstrated that increases in intracellular $[Ca^{2+}]$ cause the release of ATP through Panx1 channels (*Dahl, 2015*). Released ATP subsequently activates $Ca^{2+}$-permeable ionotropic purinergic ($P_2X$) receptors on adjacent cells to increase intracellular $Ca^{2+}$, providing a means for the intercellular propagation of $Ca^{2+}$ signals (*Barbe et al., 2006*; *Locovei et al., 2006*; *Suadicani et al., 2009*; *Vanlandewijck et al., 2018*; *Zsembery et al., 2003*). We used a pharmacological approach to determine if this mechanism was responsible for the propagation of conducted vasodilator signals in the brain capillary network. Pre-incubation of microvascular preparations with apyrase (1 U/mL) to catabolize extracellular purines (*Figure 4E*) and administration of the pan-$P_2X$ inhibitor PPADS (10 µM) blocked the dilation of upstream parenchymal arteriole segments in response to application of AITC to capillaries (*Figure 4F*). However, apyrase and PPADS had no effect on conducted vasodilation initiated by stimulation of brain capillary endothelial cell $K_{ir}$ channels with 10 mM $K^+$ (*Figure 4G and H*). Apyrase and PPADS treatment did not affect the tone of arterioles. We also found that direct application of ATP onto capillary beds stimulated conducted dilation in preparations from wild-type mice, an effect that was blocked by PPADS (*Figure 4I and J*). The velocity of ATP-conducted responses was slower than that of responses initiated by stimulation of $K_{ir}$ channels and was comparable to that of signals produced by stimulation of TRPA1 channels (*Figure 4K*). Consistent with these findings, application of ATP onto capillaries did not initiate conducted vasodilation in ex vivo preparations from *Panx1*-ecKO mice (*Figure 4L*). ATP-induced conducted vasodilation persisted in preparations from *Trpa1*-ecKO mice indicating that TRPA1 is the

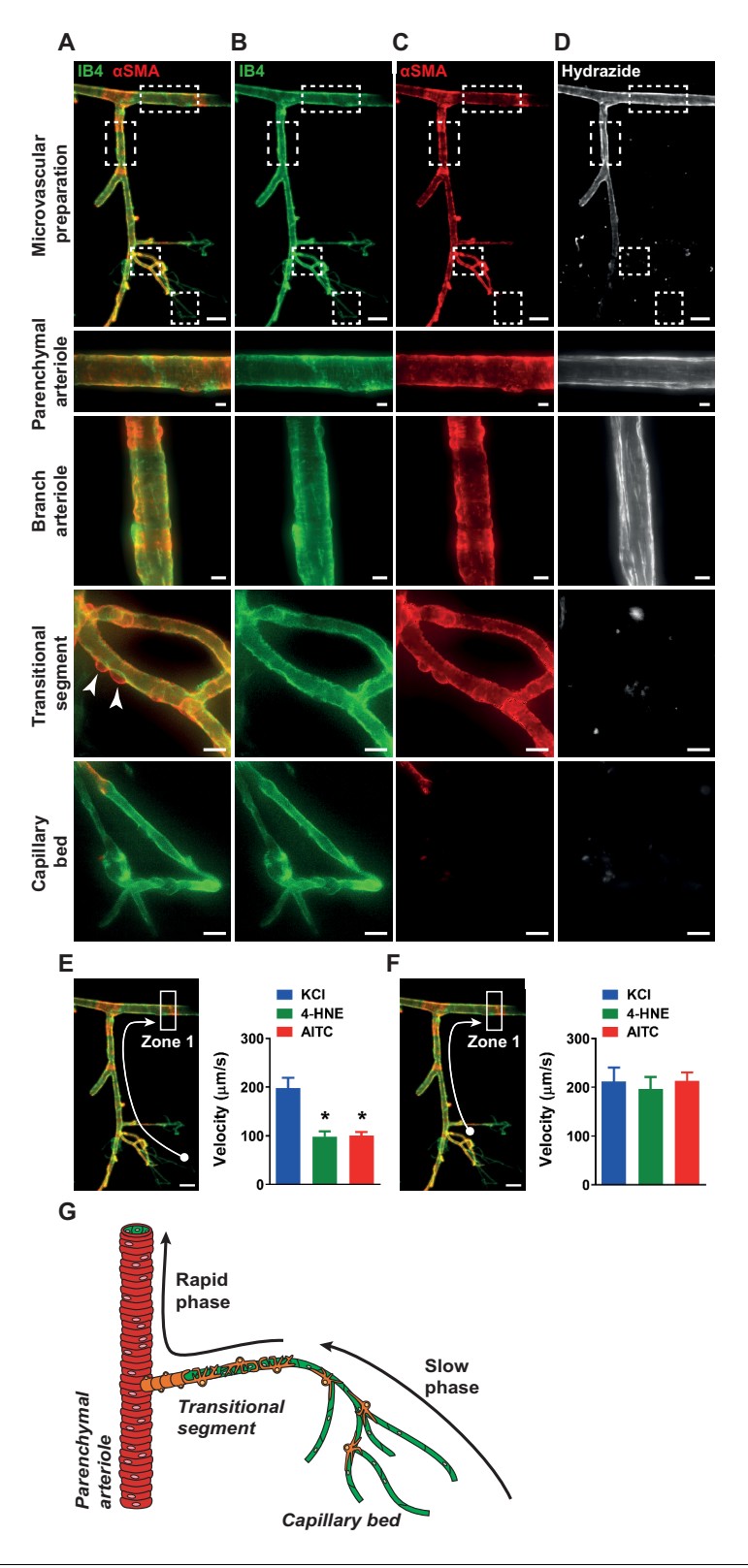

**Figure 3.** Biphasic velocity of conductive vasodilation following activation of capillary TRPA1 channels. (**A to D**) Representative fluorescence image of an ex vivo microvascular preparation obtained from a wild-type mouse. Microvascular preparations were treated with Alexa Fluor 488 conjugated isolectin B4 (IB4) (green) to label endothelial cells, and Cy3 conjugated anti-α-smooth muscle actin antibody (αSMA) (red) to label mural cells, shown merged (**A**) or separately (**B and C**). The elastin layer that is only present in arteriole segments was stained using Alexa Fluor 633 conjugated hydrazide

*Figure 3 continued on next page*

Figure 3 continued

(white) (D). Scale bar = 50 µm. Magnified images of the parenchymal arteriole, offshoot branch arteriole, post-arteriole transitional segment and capillaries showing differential mural cell coverage (white arrow) and elastin expression in these vascular segments. Scale bars = 10 µm. (E) *Left*: Conduction velocity of the vasodilator signal was calculated from the time interval and distance traveled to a specific point in the upstream arteriole (Zone 1, white arrow) following application of drugs onto capillary extremities (white ball). *Right*: Responses to the TRPA1 channel activators AITC (30 µM) and 4-HNE (1 µM) were significantly slower than the response to activation of $K_{ir}$ channels with KCl (10 mM) (n = 6–15 preparations from four to seven animals; *p<0.05, one-way ANOVA). (F). *Left:* Velocity measurements were calculated from the post-arteriole transitional segment (adjacent to the capillary bed, white ball) to a point in the upstream arteriole (Zone 1, white arrow). *Right:* Velocity following AITC (30 µM) or 4-HNE (1 µM) treatment was comparable to that following application of KCl (60 mM) (n = 6 preparations from four animals; one-way ANOVA). (G) Proposed model showing that activation of TRPA1 channels on capillaries produces a signal that slowly propagates through the capillary network and a faster propagating signal in arterioles and the post-arteriole transitional region.

The online version of this article includes the following source data for figure 3:

**Source data 1.** Individual data points and analysis summaries for datasets shown in *Figure 3*.

initiator and that Panx1-$P_2$X signaling is unaffected in these mice (*Figure 4—figure supplement 1*). These data support the concept that stimulation of TRPA1 channels on capillary endothelial cells initiates the release of ATP through endothelial cell Panx1 channels. Released ATP, in turn, stimulates $P_2$X receptors on adjacent cells to generate a propagating $Ca^{2+}$ signal.

We then asked, how far does the intercellular signal initiated by TRPA1 channels travel within the capillary network? To this end, we determined the length of cerebral capillary endothelial cells by measuring the distance between nuclei in isolated capillary networks labeled with isolectin B4 and DAPI. We found that the average capillary endothelial cell length was 28.7 ± 0.6 µm (*Figure 4—figure supplement 2A and B*). Given that the length of the capillary segment in our ex vivo microvascular preparation is in the range of 100–150 µm (*Figure 4—figure supplement 2C*), we estimate that the signal initiated by TRPA1 channels propagates through 3 to 5 endothelial cells in the capillary network before reaching post-arteriole transitional segments with mural cell coverage.

TRPA1 channels conduct mixed cation currents with a high $Ca^{2+}$ fraction (*Nilius et al., 2011*), and we have previously detected distinct TRPA1 channel-mediated $Ca^{2+}$-signaling events in cerebral endothelial cells (*Sullivan et al., 2015*). $Ca^{2+}$-imaging studies performed using microvascular preparations obtained from transgenic mice expressing the genetically encoded $Ca^{2+}$ indicator GCaMP8, exclusively in the endothelium (*Cdh5*-GCaMP8) (*Ohkura et al., 2012*) revealed that focal application of AITC onto the distal extremities of capillaries initiated an increase in intracellular [$Ca^{2+}$] (*Video 3*). Moreover, this increase in intracellular [$Ca^{2+}$] triggered similar increases in neighboring regions, indicating that the $Ca^{2+}$ signal propagates through the capillary network (*Figure 5A*). AITC-induced $Ca^{2+}$ signals were abolished by superfusing the preparation with HC-030031 (*Video 4*, *Figure 5B and C*), suggesting that initiation of the signal required the activity of TRPA1 channels. As our model predicts that the propagation velocity of the vasodilatory signal within the capillary network is slow, we therefore determined the velocity at which $Ca^{2+}$ signals travels through the capillary network. The propagation velocity of the $Ca^{2+}$ signal is defined here as the time interval between the start of an increase in intracellular [$Ca^{2+}$] at one region of interest and the start of an increase in intracellular [$Ca^{2+}$] in a neighboring region, normalized to the distance between the two regions. As expected, the velocity of vasodilator $Ca^{2+}$ signals within the capillaries travel at speeds slower than the rapid vasodilator signal traveling through vascular segments covered by mural cells (*Figure 5D*). Focal application of ATP was also found to initiate an increase in intracellular [$Ca^{2+}$], that propagates to adjacent cells (*Video 5*, *Figure 5E*). PPADS abolished ATP-induced $Ca^{2+}$ signal, suggesting that initiation and propagation of $Ca^{2+}$ signals require $P_2$X receptors (*Video 6*, *Figure 5F and G*). The velocity of ATP-induced $Ca^{2+}$ signals is similar to that produced by activating TRPA1 channels, suggesting a convergent vasodilator signaling pathway (*Figure 5H*). The duration, rise time and decay time of the $Ca^{2+}$ signal evoked by AITC and ATP were also comparable (*Figure 5—figure supplement 1A* to C). We next validated that the vasodilator signal through the capillaries requires influx of extracellular $Ca^{2+}$ through purinergic $P_2$X receptors by superfusing preparations with aCSF devoid of extracellular $Ca^{2+}$. We found that ATP-induced $Ca^{2+}$ signals were abolished in preparations under 0 $Ca^{2+}$ conditions; however, the response returned once extracellular $Ca^{2+}$ (2 mM) was reintroduced (*Videos 7–9*, *Figure 5I and J*). This suggests that the propagative $Ca^{2+}$ signal is dependent on

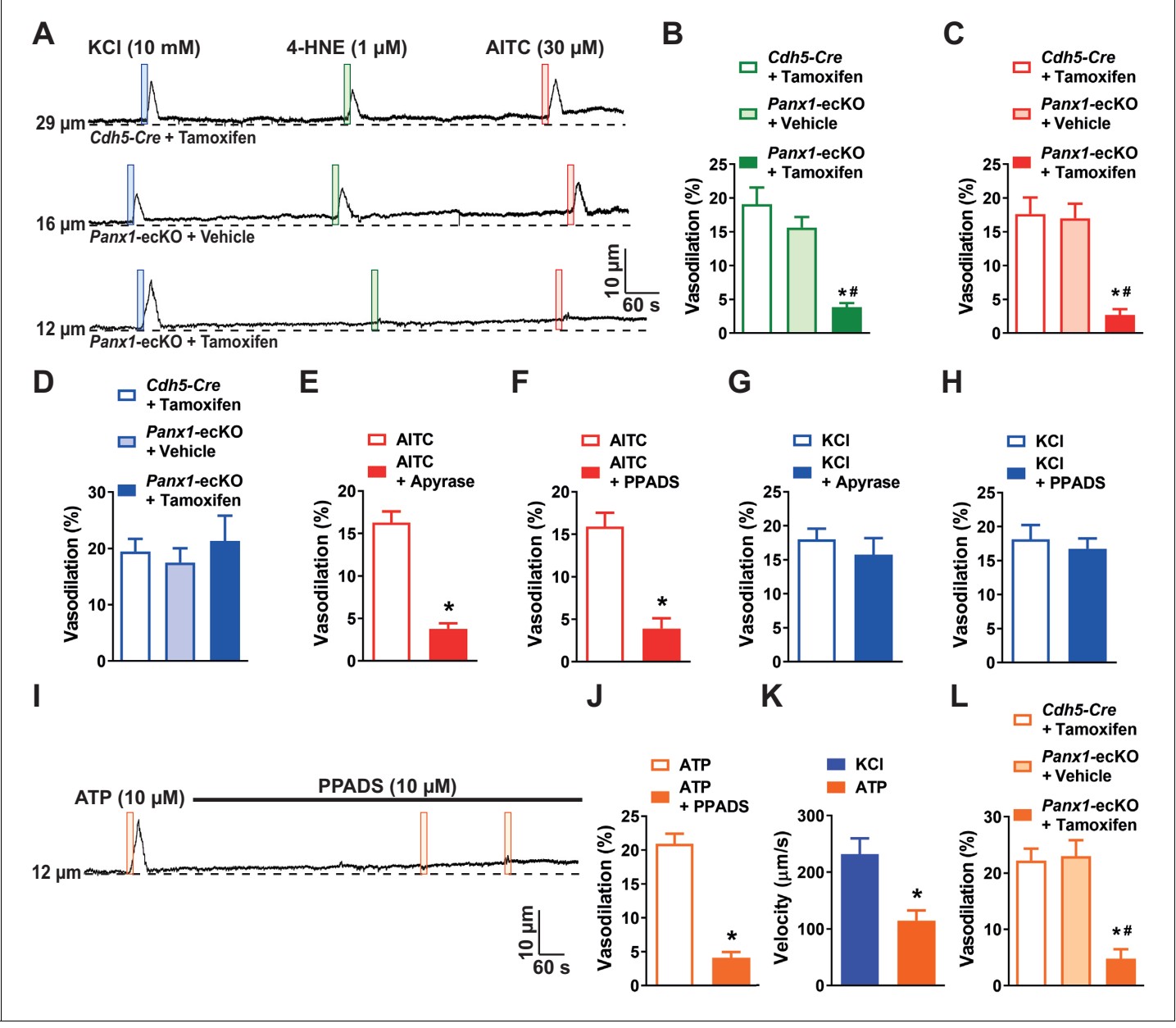

**Figure 4.** Activation of capillary TRPA1 channels produces a purinergic signal that travels through the capillary bed. (A) Representative traces showing that application of AITC (30 µM; red box) or 4-HNE (1 µM; green box) onto capillary extremities did not dilate the upstream arteriole in microvascular preparations from *Panx1*-ecKO mice, whereas vasodilatory effects of elevated KCl (10 mM; blue box) were unchanged. (B and C) Summary data showing that neither 4-HNE (1 µM) (B) nor AITC (30 µM) (C) evoked dilation of upstream arterioles in preparations from *Panx1*-ecKO mice (n = 6 preparations from five animals; *p<0.05 vs. *Cdh5-Cre* injected with tamoxifen, #p<0.05 vs. *Panx1*-ecKO mice injected with vehicle (peanut oil), one-way ANOVA. (D) Summary data showing that the response to elevated KCl (10 nM) was unchanged in preparations from *Panx1*-ecKO mice (n = 6 preparations from five animals; one-way ANOVA). (E) Summary data showing that catabolism of extracellular purines with apyrase (1 U/mL) blunted the response to AITC (30 µM) in microvascular preparations from wild-type mice (n = 9 preparations from eight animals; *p<0.05, paired *t*-test). (F) Summary data showing that the pan-P2X inhibitor PPADS (10 µM) inhibited the response to AITC (30 µM) (n = 9 preparations from six animals; *p<0.05, paired *t*-test). (G and H). The response to elevated KCl (10 mM) was unaffected by treatment with apyrase (1 U/mL) (G) or PPADS (10 µM) (H) (n = 9 preparations from six to eight animals; paired *t*-test). (I) Representative trace showing that application of ATP (10 µM; orange box) onto capillary extremities dilated the upstream arteriole in microvascular preparations from wild-type mice, and that this response was blocked by the pan-P2X inhibitor PPADS (10 µM). (J) Summary data showing that PPADS (10 µM) inhibited the response ATP (10 µM) responses (n = 6 preparations from four animals; *p<0.05, paired *t*-test). (K) The velocity of the ATP (10 µM) response was significantly slower compared with that to activation of Kir channels with KCl (60 mM) (n = 6 preparations from four animals; *p<0.05, paired *t*-test). (L) Summary data showing that ATP (10 µM) did not evoke dilation of

*Figure 4 continued on next page*

*Figure 4 continued*

upstream arterioles in preparations from *Panx1*-ecKO mice (n = 6 preparations from three to four animals; *p<0.05 vs. *Cdh5-Cre* injected with tamoxifen, #p<0.05 vs. *Panx1*-ecKO mice injected with vehicle (peanut oil), one-way ANOVA).

The online version of this article includes the following source data and figure supplement(s) for figure 4:

**Source data 1.** Individual data points and analysis summaries for datasets shown in *Figure 4*.

**Figure supplement 1.** ATP-induced dilation persisted in preparations from *Trpa1*-ecKO mice.

**Figure supplement 1—source data 1.** Individual data points and analysis summaries for datasets shown in *Figure 4—figure supplement 1*.

**Figure supplement 2.** Average length of a cerebral capillary endothelial cell.

**Figure supplement 2—source data 1.** Individual data points and analysis summaries for datasets shown in *Figure 4—figure supplement 2*.

ionotropic purinergic $P_2X$ receptor, and not through the release of $Ca^{2+}$ from intracellular stores downstream of G protein-coupled $P_2Y$ receptor activation.

Taken together, these data support a model in which slow, short-range intercellular $Ca^{2+}$ signals propagate through capillary endothelial cells via a mechanism that is dependent on ATP release through Panx1 channels and activation of purinergic $P_2X$ receptors on adjacent endothelial cells (*Figure 5K*). This pathway is necessary for the dilation of upstream arterioles in response to stimulation of TRPA1 channels in capillary endothelial cells.

## Rapid-phase propagation of vasodilator signals is initiated in the post-arteriole transitional region and requires $K_{ir}$ channel activity

We next turned to the mechanisms responsible for the rapid phase of the propagating vasodilator signal. Because signal propagation velocities were the same in post-arteriole transitional segments and parenchymal arterioles, regardless of the initiating stimulus, we hypothesized that propagation in this segment occurs by rapid electrical communication, as previously described by *Longden et al., 2017*. To test this, we performed experiments using a modified capillary-arteriole microvascular preparation in which the capillary tree was removed while leaving the post-arteriole transitional segment intact (*Figure 6A*). In control experiments, we found that Evans Blue dye ejected from a picospritzer near the post-arteriole transitional segment did not spread to the upstream parenchymal arteriole (*Video 10*), supporting our ability to selectively stimulate the transitional segments.

Interestingly, we found that focal application of elevated [$K^+$], 4-HNE, or AITC onto the post-arteriole transitional segment caused dilation of the upstream parenchymal arteriole (*Figure 6B and C*), indicating that functional $K_{ir}$ and TRPA1 channels are present in these transitional segments and are capable of producing conducted vasodilator responses when stimulated. The propagation velocities of the conducted vasodilator signals produced by stimulating post-arteriole transitional segment with elevated [$K^+$], AITC, or 4-HNE were nearly identical, suggesting a common signal-propagation mechanism (*Figure 6D*). We also found that vasodilator responses to elevated [$K^+$] (*Figure 6E*) and AITC (*Figure 6F*) were blocked by inhibiting $K_{ir}$ channels with $BaCl_2$. The rapid signal-propagation velocity and sensitivity to blockade by $BaCl_2$ suggest that fast electrical communication involving $K_{ir}$ channels (*Longden et al., 2017*) is a convergent means of vasodilator signal conduction in vascular segments with mural cell coverage. We also found that superfusing the ex vivo preparation with $BaCl_2$ significantly blunted the

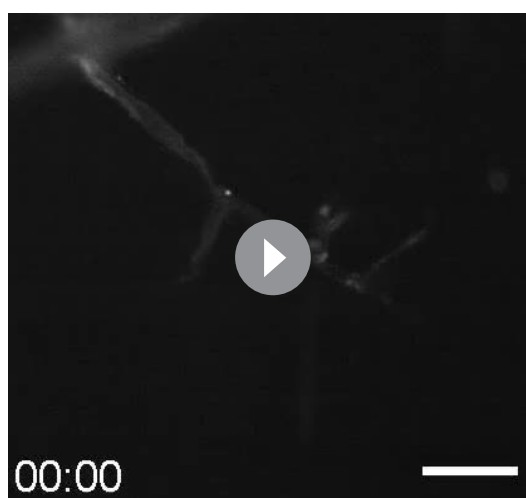

**Video 3.** Focal application of AITC onto capillary extremities induces an increase in intracellular [$Ca^{2+}$]. Representative time-series images of a microvascular preparation demonstrating that localized application of AITC (30 µM) onto distal capillary extremities produces a propagative $Ca^{2+}$ signal. AITC was applied after 10 s. Scale bar = 50 µm.
https://elifesciences.org/articles/63040#video3

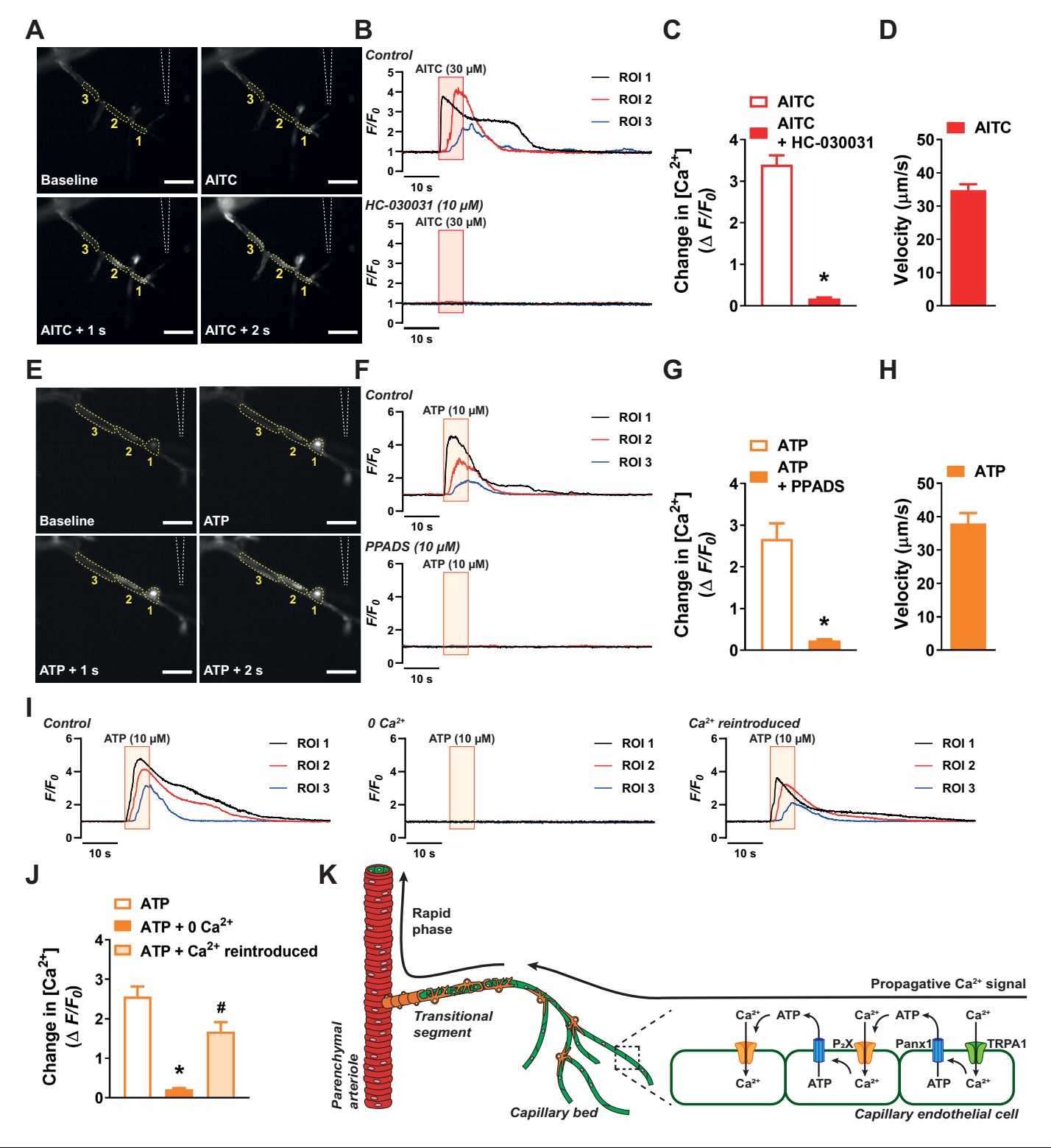

**Figure 5.** Activation of capillary TRPA1 channels and purinergic receptors produces a $Ca^{2+}$ response that travels through the capillary bed. (**A**) Representative time course images demonstrating the fractional increase in fluorescence ($F/F_0$) of the $Ca^{2+}$ signal in a capillary segment in microvascular preparations from transgenic mice expressing a genetically encoded $Ca^{2+}$ indicator, GCaMP8, exclusively in the endothelium (*Cdh5*-GCaMP8). The tip of the micropipette (outlined) was placed adjacent to distal extremities of capillaries. Focal application of AITC (30 μM) evoked a propagative $Ca^{2+}$ signal that was observed in adjacent region of interests. Scale bar = 50 μm. (**B**) Representative traces showing the fractional increase

*Figure 5 continued on next page*

*Figure 5 continued*

in florescence following application of AITC (30 µM; red box) that was blocked by the selective TRPA1 antagonist HC-030031 (10 µM). (C) Summary data showing that HC-030031 (10 µM) inhibited the response to AITC (30 µM) (n = 6 preparations from five animals; *p<0.05, paired *t*-test). (D) The velocity of the response through capillary segments evoked by activation of TRPA1 channels (n = 6 preparations from five animals). (E) Representative time course images demonstrating the fractional increase in fluorescence of the $Ca^{2+}$ signal produced following focal application of ATP (10 µM) to distal capillaries. Propagation of the $Ca^{2+}$ signal was observed in the adjacent region of interests following ATP (10 µM) application. Scale bar = 50 µm. (F) Representative traces showing the fractional increase in florescence following application of ATP (10 µM; orange box) that was blocked by the pan-$P_2X$ inhibitor PPADS (10 µM). (G) Summary data showing that PPADS (10 µM) inhibited the response to ATP (10 µM) (n = 7 preparations from seven animals; *p<0.05, paired *t*-test). (H) The velocity of the response through capillary segments evoked by activation of purinergic receptors (n = 13 preparations from 12 animals). (I) Representative traces showing the fractional increase in florescence following application of ATP (10 µM; orange box) was abolished when the preparation was superfused in $Ca^{2+}$-free aCSF. The response was restored following the reintroduction of extracellular $Ca^{2+}$. (J) Summary data showing that bathing the preparation in $Ca^{2+}$-free aCSF solution attenuated the response to ATP (10 µM), but was restored once $Ca^{2+}$ (2 mM) was reintroduced (n = 6 preparations from five animals; *p<0.05 vs. control ATP response, #p<0.05 vs. preparations bathed in 0 $Ca^{2+}$, one-way ANOVA). (K) Illustration of the proposed model for signal propagation through the capillary bed. Increases in intracellular $[Ca^{2+}]$ caused by TRPA1 channel-mediated $Ca^{2+}$ influx induce ATP release through Panx1 channels, which in turn activates purinergic $P_2X$ receptors on the adjacent endothelial cell.

The online version of this article includes the following source data and figure supplement(s) for figure 5:

**Source data 1.** Individual data points and analysis summaries for datasets shown in *Figure 5*.

**Figure supplement 1.** Kinetics of the $Ca^{2+}$ response in capillaries following activation of TRPA1 channels purinergic receptors.

**Figure supplement 1—source data 1.** Individual data points and analysis summaries for datasets shown in *Figure 5—figure supplement 1*.

response to localized application of AITC onto capillary extremities (*Figure 6G*). Interestingly, focal application of ATP onto the post-arteriole transitional segment did not evoke dilation of upstream arterioles (*Figure 6—figure supplement 1A and B*), suggesting that ATP-dependent propagation must be initiated at more distal points in the capillary network.

We next investigated how the slowly propagating $Ca^{2+}$ signals initiated by capillary endothelial cell TRPA1 channels are converted to rapidly propagating electrical signals in mural cell-covered post-arteriole transitional segments to cause dilation of upstream parenchymal arterioles. Combined inhibition of nitric oxide synthase with L-NAME (100 µM) and cyclooxygenase with indomethacin (10 µM) in microvascular preparations had no effect on arteriole dilation induced by stimulation of capillary extremities with AITC (*Figure 6H*). However, inhibition of IK and SK channel activity by superfusing TRAM34 (1 µM) and apamin (300 nM), respectively, significantly blunted upstream dilation in response to focal stimulation of capillary endothelial cell TRPA1 channels (*Figure 6I*), demonstrating that IK and SK channel activity is necessary for the conducted vasodilator responses initiated by brain capillary endothelial cell TRPA1 channels. Although it has been reported that functional IK and SK channels are not present in capillary endothelial cells (*Longden et al., 2017*; *Figure 1—figure supplement 1*), our functional data suggest that IK and SK channels are expressed in post-arteriole transitional segments, where they convert slowly propagating intercellular $Ca^{2+}$ signals initiated by capillary TRPA1 channels into rapidly propagating electrical signals. Direct application of NS309 (10 µM) directly onto parenchymal arterioles, which are known to express endothelial IK and SK channels caused dilation, whereas focal stimulation of

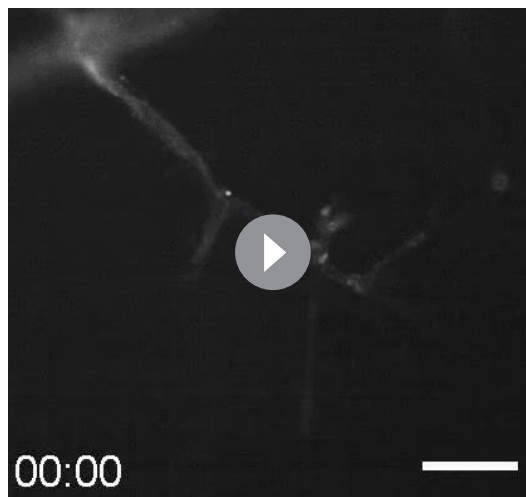

**Video 4.** Increase in intracellular $[Ca^{2+}]$ initiated by AITC is blocked by the selective TRPA1 antagonist HC-030031. Representative time-series images of a microvascular preparation demonstrating that the propagative $Ca^{2+}$ signal produced by AITC (30 µM) is blocked by superfusing the preparation with HC-030031 (10 µM). AITC was applied after 10 s. Scale bar = 50 µm.
https://elifesciences.org/articles/63040#video4

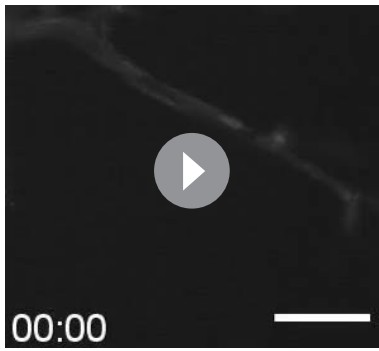

**Video 5.** Focal application of ATP onto capillary extremities induces an increase in intracellular [Ca$^{2+}$]. Representative time-series images of a microvascular preparation demonstrating that localized application of ATP (10 µM) onto distal capillary extremities produces a propagative Ca$^{2+}$ signal. ATP was applied after 10 s. Scale bar = 50 µm.
https://elifesciences.org/articles/63040#video5

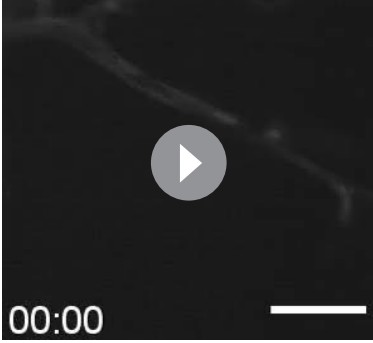

**Video 6.** Increase in intracellular [Ca$^{2+}$] initiated by AITC is blocked by the pan-P2X inhibitor PPADS. Representative time-series images of a microvascular preparation demonstrating that the propagative Ca$^{2+}$ signal produced by ATP (10 µM) is blocked by superfusing the preparation with PPADS (10 µM). ATP was applied after 10 s. Scale bar = 50 µm.
https://elifesciences.org/articles/63040#video6

capillary extremities did not (*Figure 6—figure supplement 2A and B*). Surprisingly, we found that focal application of NS309 directly onto the post-arteriole transitional segment induced dilation of the upstream parenchymal arteriole, indicating that IK and SK channels are expressed in this region and that their activation results in conducted vasodilation (*Figure 6J and K*). We also found that this response was inhibited by BaCl$_2$ (*Figure 6K*) and propagated at the same velocity as signals initiated by stimulating the post-arteriole transitional segment with elevated [K$^+$] and TRPA1 agonists (*Figure 6L*). These data support the concept that the Ca$^{2+}$ signals arriving from the capillary network activate IK and SK channels within the post-arteriole transitional segment to initiate a fast-conducting vasodilator response that requires K$_{ir}$ channel activity (*Figure 6M*).

## Stimulation of brain endothelial cell TRPA1 channels increases local blood flow in vivo

We next performed experiments to determine if stimulation of endothelial cell TRPA1 channels increase localized cerebral blood flow in vivo. Red blood cell (RBC) flux within capillaries and changes in diameter of upstream arterioles were visualized through a cranial window using two-photon laser-scanning microscopy (*Figure 7A*). Fluorescein isothiocyanate (FITC)-conjugated dextran (*i. v.*) was administered to animals to allow visualization of the vasculature and contrast imaging of RBCs. TRPA1 channels were locally activated by pressure ejecting AITC directly onto a single capillary using a micropipette. In control *Trpa1$^{fl/fl}$* animals, local application of AITC (30 µM) onto a capillary produced a significant increase in RBC flux within the stimulated capillary (*Figure 7B and C*). The rate of flux increase was 11.6 ± 2.3 RBC/s (*Figure 7—figure supplement 1A*). The latency of this response (7.5 ± 1.2 s, *Figure 7—figure supplement 1B*) was slow compared with the response induced by applying elevated [K$^+$] (3.8 ± 0.9 s), as reported by *Longden et al., 2017*. This finding is consistent with the slower kinetics of the TRPA1-mediated vasodilator response compared with the K$_{ir}$-mediated response observed in our ex vivo microvascular preparations (*Figure 3A*). In contrast, AITC failed to elicit an increase in RBC flux

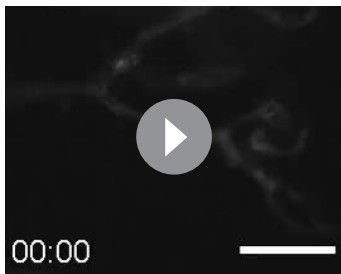

**Video 7.** ATP-induced Ca$^{2+}$ signal under normal conditions. Representative time-series images of a microvascular preparation demonstrating that localized application of ATP (10 µM) onto distal capillary extremities produces a propagative Ca$^{2+}$ signal in preparations superfused with Ca$^{2+}$-containing aCSF. ATP was applied after 10 s. Scale bar = 50 µm.
https://elifesciences.org/articles/63040#video7

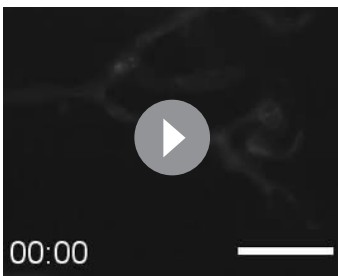

**Video 8.** ATP-induced Ca²⁺ signal is abolished in extracellular Ca²⁺-free conditions. Representative time-series images of a microvascular preparation demonstrating that localized application of ATP (10 µM) onto distal capillary extremities failed to induced a propagative Ca²⁺ signal in preparations superfused with Ca²⁺-free aCSF. ATP was applied after 10 s. Scale bar = 50 µm.
https://elifesciences.org/articles/63040#video8

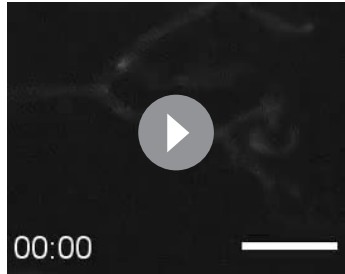

**Video 9.** ATP-induced Ca²⁺ signal is restored following reintroduction of extracellular Ca²⁺. Representative time-series images of a microvascular preparation demonstrating that the propagative Ca²⁺ signal following localized application of ATP (10 µM) onto distal capillary extremities returned once extracellular Ca²⁺ (2 mM) was reintroduced to the preparation. ATP was applied after 10 s. Scale bar = 50 µm.
https://elifesciences.org/articles/63040#video9

in almost all (6 of 7) *Trpa1*-ecKO mice tested (*Figure 7D* to F). We also confirmed vasodilatory responses to capillary stimulation by assessing changes in the cross-sectional area of the feeding arteriole, demonstrating that local application of AITC onto a capillary significantly increased the area of the upstream arteriole in *Trpa1^fl/fl^* animals (*Figure 7G*). Furthermore, the increase in cross-sectional area observed in arterioles of *Trpa1^fl/fl^* animals was greater than that observed in *Trpa1*-ecKO mice (*Trpa1^fl/fl^* mice 72.2 ± 10.8 µm² (n = 11) vs. *Trpa1*-ecKO mice 8.3 ± 3.9 µm² (n = 7), p<0.05) (*Figure 7H and I*). These data indicate that activation of endothelial cell TRPA1 channels in the brain dilates upstream arterioles and increases RBC flux in capillaries.

## Functional hyperemia in the somatosensory cortex requires endothelial cell TRPA1 channels

To assess the role of TRPA1 channels in functional hyperemia in vivo, we measured relative changes in blood flow in the somatosensory cortex using a thinned-skull mouse model (*Figure 8—figure supplement 1*). Changes in blood flow were induced by contralateral whisker stimulation and were recorded using laser-Doppler flowmetry (*Girouard et al., 2010*; *Park et al., 2014*). Stimulating whiskers for 5 s reliably and reproducibly increased cerebral blood flow in wild-type mice (*Figure 8B*). Control experiments indicated that this was not due to artifactual noise, as stimulation of ipsilateral whiskers was without effect (*Figure 8—figure supplement 2*). To determine if TRPA1 channels are involved in this functional hyperemic response, we treated mice with HC-030031 (100 mg/kg, *i.p.* for 30 min). HC-030031 treatment significantly attenuated the increase in cerebral blood flow following whisker stimulation (*Figure 8A and B*). Our model suggests that TRPA1 channels should affect the slow components of the hyperemic response. Further analysis of the kinetics revealed that the duration of the response was reduced (*Figure 8C*), and the decay rate was increased (*Figure 8D*) following HC-030031 treatment, suggesting that TRPA1 channels are needed to sustain the rise in blood flow. The rise rate and latency of the response was unaffected (*Figure 8E and F*). To demonstrate the contribution of endothelial cell TRPA1 channels, we assessed functional hyperemia in *Trpa1*-ecKO mice. The basal increase in blood flow following 5 s whisker stimulation was significantly blunted in *Trpa1*-ecKO mice compared with *Trpa1^fl/fl^* mice (*Figure 8G and H*). Similarly, only the duration and decay rate were found to be impaired in *Trpa1*-ecKO mice (*Figure 8I–L*). Collectively, these data demonstrate that endothelial cell TRPA1 channels contribute to the increase in blood flow and sustains that vasodilatory response. Interestingly, TRPA1 channels were found not to be involved when contralateral whiskers were stimulated for a shorter duration. The increase in blood flow and the kinetics of the hyperemic response were unchanged in animals treated with HC-030031 (*Figure 8M–R*) and *Trpa1*-ecKO mice (*Figure 8S–X*) following either a 1 or 2 s (*Figure 8—figure supplement 3*) stimulation period. Our prior two-photon data indicated an approximate 7 s delay before TRPA1-dependent increase in blood flow (*Figure 7—figure supplement 1B*). It is

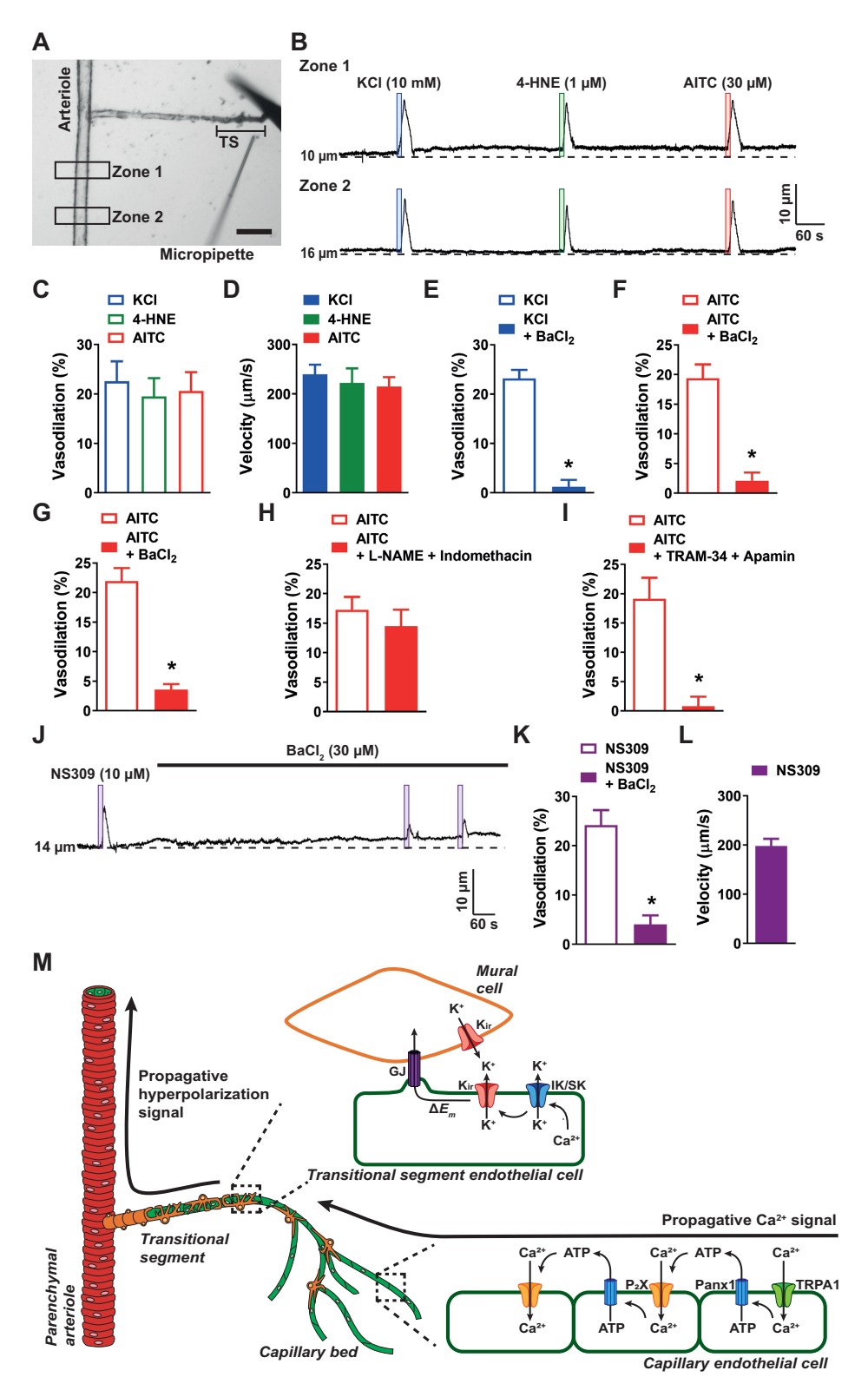

**Figure 6.** Rapid propagation of TRPA1-induced vasodilation is dependent on $K_{ir}$, IK, and SK channels. (**A**) Representative image of a cerebral microvascular preparation in which the capillary bed was removed while leaving the post-arteriole transitional segment (TS) intact. The transitional segment is located at the distal end of the offshoot arteriole branch. Scale bar = 100 μm. A picospritzing cannula was used to directly apply drugs to the post-arteriole transitional segment. (**B**) Representative traces showing that application of elevated KCl (10 mM; blue box), AITC (30 μM; red box), or

*Figure 6 continued on next page*

*Figure 6 continued*

4-HNE (1 µM; green box) onto the post-arteriole transitional segment increased the lumen diameter of the upstream arteriole in microvascular preparations from wild-type mice. (C) Summary data showing dilation produced by elevated KCl (10 mM), AITC (30 µM) and 4-HNE (1 µM) (n = 6–8 preparations from four to five animals; one-way ANOVA). (D) Velocity was calculated from the post-arteriole transitional segment to a point in the upstream arteriole following localized drug application onto the post-arteriole transitional segment. Velocity measurements were comparable for elevated KCl (10 mM), 4-HNE (1 µM), and AITC (30 µM) (n = 6–8 preparations from four to five animals; one-way ANOVA). (E and F) Summary data showing that BaCl$_2$ (30 µM) significantly reduced the response to elevated KCl (10 mM) (E) and AITC (30 µM) (F) when applied to the post-arteriole transitional segment (n = 6 preparations from four animals; *p<0.05, paired *t*-test). (G) Summary data showing that BaCl$_2$ (30 µM) significantly attenuated the response to AITC (30 µM) when directly applied onto capillary extremities (n = 6 preparations from four animals; *p<0.05, paired *t*-test). (H) Summary data showing that combined inhibition of nitric oxide synthase and cyclooxygenase with L-NAME (100 µM) and indomethacin (10 µM), respectively, did not affect the response to AITC (30 µM) when directly onto capillary extremities (n = 7 preparations from four animals; paired *t*-test). (I) Summary data showing that combined IK and SK channel inhibition with TRAM-34 (1 µM) and apamin (300 nM), respectively, significantly attenuated the response to AITC (30 µM) when directly applied onto capillary extremities (n = 7 preparations from five animals; *p<0.05, paired *t*-test). (J and K) Representative trace (J) and summary data (K) showing that application of the IK and SK channel activator NS309 (10 µM; purple box) directly onto the post-arteriole transitional segment dilated the upstream arteriole, and that this response was attenuated by inhibition of K$_{ir}$ channels with BaCl$_2$ (30 µM) (n = 6 preparations from four animals; *p<0.05, paired *t*-test). (L) The velocity of the response produced by application of the IK and SK channel activator NS309 (10 µM) onto the post-arteriole transitional segment was similar to that produced by activation of K$_{ir}$ channels or TRPA1 channels (n = 6 preparations from four animals). (M) Illustration of the proposed signaling model depicting events following activation of capillary endothelial TRPA1 channels. Once the Ca$^{2+}$ signal from the capillary bed arrives at the post-arteriole transitional segment, IK and SK channels are activated and facilitate K$^+$ efflux. This in turn activates K$_{ir}$ channels, which propagate the signal retrogradely to cause dilation of the upstream arteriole.

The online version of this article includes the following source data and figure supplement(s) for figure 6:

**Source data 1.** Individual data points and analysis summaries for datasets shown in *Figure 6*.
**Figure supplement 1.** ATP does not evoke dilation of upstream arterioles when applied to the post-arteriole transitional segment.
**Figure supplement 1—source data 1.** Individual data points and analysis summaries for datasets shown in *Figure 6—figure supplement 1*.
**Figure supplement 2.** Application of NS309 to capillary extremities has no effect on the upstream arteriole.
**Figure supplement 2—source data 1.** Individual data points and analysis summaries for datasets shown in *Figure 6—figure supplement 2*.

therefore unlikely that this slow propagation pathway contributes to the hyperemic response under shorter stimulation. However, during extended periods of neuronal activation, the slowly propagating TRPA1 pathway is necessary for functional hyperemia in the somatosensory cortex of the brain.

## Discussion

The parenchyma of the brain is exposed to an ever-changing milieu of physical, environmental, endocrine, paracrine, metabolic, and neurochemical stimuli. The neurovascular apparatus must detect and decode signals generated by active areas to properly direct regional blood flow and ensure optimal brain function. The preponderance of the brain vasculature is composed of densely packed capillary networks; an optical volumetric analysis has revealed that the soma of all central neurons are located no more than 15 µm from a brain capillary endothelial cell (*Tsai and Kleinfeld, 2009*). Building on these observations, Longden et al. introduced a convincing new paradigm for NVC, demonstrating that conducted vasodilator responses orchestrated by capillary endothelial cells provoke functional hyperemia in the brain through propagating electrical (hyperpolarizing) signals initiated and sustained by K$_{ir}$ channels. Thus, the brain's extensive capillary network serves as a sensory web for detecting [K$^+$] released at active neuronal synapses (*Longden et al., 2017*). The

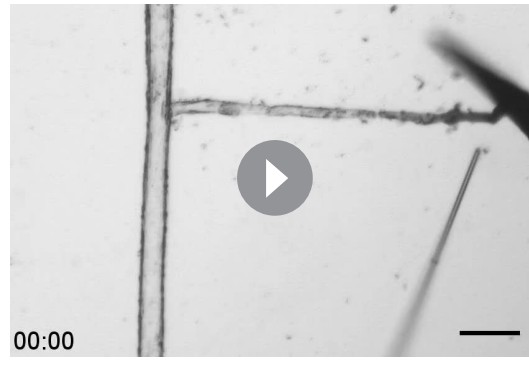

**Video 10.** Localized application of Evans Blue dye onto the post-arteriole transitional segment of a microvascular preparation. Representative time-series images of a modified microvascular preparation in which the capillary tree was removed. Application of Evans Blue dye (1% w/v) onto the post-arteriole transitional segment was localized to this region and did not spread to the upstream parenchymal arteriole. Evans Blue was applied after 10 s. Scale bar = 100 µm.
https://elifesciences.org/articles/63040#video10

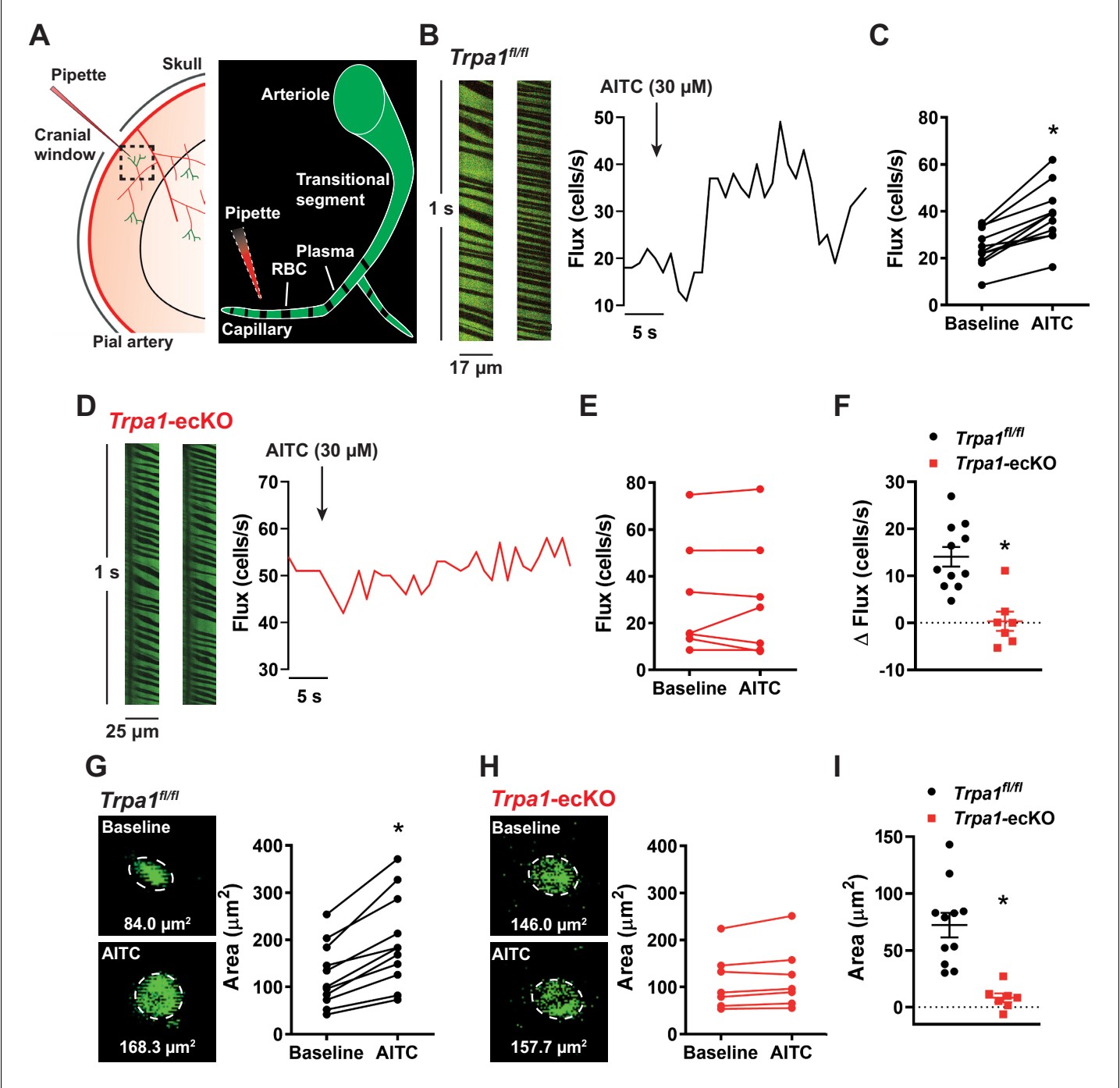

**Figure 7.** In vivo stimulation of capillary TRPA1 channels increases RBC flux within the capillary bed. (**A**) Experimental illustration. Mice were injected with FITC-conjugated dextran to allow visualization of the cortical vasculature through a cranial window using in vivo two-photon laser-scanning microscopy. RBCs appear as black shadows against green fluorescent plasma. A pipette containing AITC and TRITC-dextran (for pipette visualization) was positioned next to a capillary. (**B**) RBC flux through a single capillary was examined by analyzing distance-time plots of capillary line scans. TRPA1 channels on capillary endothelial cells were locally stimulated by picospritzing AITC (30 µM) directly onto a single capillary. Representative line-scan plots of a capillary (*left*) and representative time-flux trace (*right*) demonstrating changes in RBC flux before and after application of AITC (30 µM) in control *Trpa1*^fl/fl mice. (**C**) Summary data showing that AITC (30 µM) increased RBC flux in the stimulated capillary of *Trpa1*^fl/fl mice (n = 11 animals; *p<0.05, paired *t*-test). (**D**) Representative line-scan plots of a capillary (*left*) and representative time-flux trace (*right*) demonstrating that application of AITC (30 µM) had no effect in *Trpa1*-ecKO mice. (**E**) Summary data showing that AITC (30 µM) had no effect in *Trpa1*-ecKO mice (n = 7 animals, paired *t*-test). (**F**) Change in RBC flux in control *Trpa1*^fl/fl versus *Trpa1*-ecKO mice (n = 7–11 animals; *p<0.05, unpaired *t*-test). (**G and H**) Representative images and summary data showing change in cross-sectional area of the upstream arteriole following application of AITC (30 µM) onto a capillary of control

*Figure 7 continued on next page*

*Figure 7 continued*

*Trpa1$^{fl/fl}$* (**G**) and *Trpa1*-ecKO (**H**) mice (n = 7–11 animals; *p<0.05, paired *t*-test). (**I**) Change in arteriole cross-sectional area was less in *Trpa1*-ecKO mice compared to control *Trpa1$^{fl/fl}$* (n = 7–11 animals; *p<0.05, unpaired *t*-test).

The online version of this article includes the following source data and figure supplement(s) for figure 7:

**Source data 1.** Individual data points and analysis summaries for datasets shown in *Figure 7*.
**Figure supplement 1.** Kinetics of the RBC flux increase following in vivo stimulation of capillary TRPA1 channels.
**Figure supplement 1—source data 1.** Individual data points and analysis summaries for datasets shown in *Figure 7—figure supplement 1*.

NVC response is essential for life itself, and the multiplicity of distinct signaling modalities in the active brain imply that brain capillary endothelial cells possess an equally broad assortment of complementary and overlapping sensors. Here, we investigated this concept, providing evidence that activation of brain capillary endothelial cell TRPA1 channels promotes dilation of upstream parenchymal arterioles through a novel, biphasic intercellular signaling mechanism that is dependent on Panx1 channels. Our findings also identify purinergic receptors on capillary endothelial cells as part of this signal conduction pathway and as independent detectors of released ATP. We further demonstrate that focal stimulation of $K_{ir}$ and TRPA1 channels in the post-arteriole transitional region results in conducted dilation of upstream arterioles, expanding the brain's vascular sensory network to include these segments. We conclude that multiple molecular sensors that are critically important for NVC and functional hyperemia in the brain, including TRPA1 channels and purinergic receptors, are present in brain capillary endothelial cells.

Our findings demonstrate that capillary endothelial cell TRPA1 channels initiate signals that propagate through the cerebral microvasculature to cause dilation of upstream parenchymal arterioles, locally increase RBC flux, and produce functional hyperemia in the brain. TRPA1 exhibits a unique pattern of expression within the vasculature: it is present in the endothelium of arteries and arterioles in the brain, but not in other organs, such as the heart, dermis, kidney, or mesentery (*Earley et al., 2009*; *Pires and Earley, 2018*; *Sullivan et al., 2015*). Activation of vascular TRPA1 channels produces endothelium-dependent dilation of cerebral arteries and arterioles, a mechanism that is neuroprotective during ischemic stroke (*Earley et al., 2009*; *Pires and Earley, 2018*; *Sullivan et al., 2015*). Lipid peroxidation metabolites generated by reactive oxygen species (ROS), such as 4-HNE, directly activate TRPA1 channels (*Andersson et al., 2008*; *Taylor-Clark et al., 2008*). We have shown that superoxide anions ($O_2^-$), generated by the activity of NADPH oxidase 2 (NOX2) or by mitochondrial respiration under hypoxic and ischemic conditions, endogenously generate 4-HNE, which stimulates TRPA1 channel activity in the cerebral endothelium to cause vasodilation (*Pires and Earley, 2018*; *Sullivan et al., 2015*). We propose that extracellular ROS generated by astrocytes could act as endogenous agonists of capillary endothelial cell TRPA1 channels during NVC. One possible source of ROS is astrocytic endfoot processes, which encase brain capillaries and have been shown to express NOX2 and produce ROS (*Abramov and Duchen, 2005*). Alternatively, extracellular ROS generated by metabolically active neurons proximal to brain capillaries could activate TRPA1 channels. Pericytes and microglia cells are also intimately associated with capillaries in the brain and are potential sources of TRPA1-activating ROS metabolites.

The vasodilator signals initiated by brain capillary endothelial cell TRPA1 channels propagate at about half the speed of those initiated by $K_{ir}$, suggesting that the two signals are conducted by different pathways. Prior work has demonstrated that activation of capillary endothelial cell $K_{ir}$ channels generates a fast-conducting electrical signal that hyperpolarizes smooth muscle cells in upstream arterioles to cause vasodilation (*Longden et al., 2017*). Our data showed that stimulation of capillary endothelial cell TRPA1 channels produced slowly propagating, short-range intercellular signals that are dependent on endothelial expression of Panx1 channels. Panx1 forms transmembrane channels in cerebral arteriole and capillary endothelial cells (*Vanlandewijck et al., 2018*). Panx1 channels are normally closed, but when activated by increases in intracellular [$Ca^{2+}$], they release ATP (*Dahl, 2015*; *Locovei et al., 2006*). Previous studies using exogenous expression systems have shown that Panx1 channels and $Ca^{2+}$-permeable purinergic $P_2X$ receptors colocalize on the membrane of *Xenopus* oocytes (*Locovei et al., 2007*) and urothelial cells (*Negoro et al., 2014*) to form signaling microdomains that permit intercellular propagation of $Ca^{2+}$ signals. In this mechanism, ATP released via Panx1 channels binds and activates $P_2X$ receptors on neighboring cells, resulting in $Ca^{2+}$ influx and

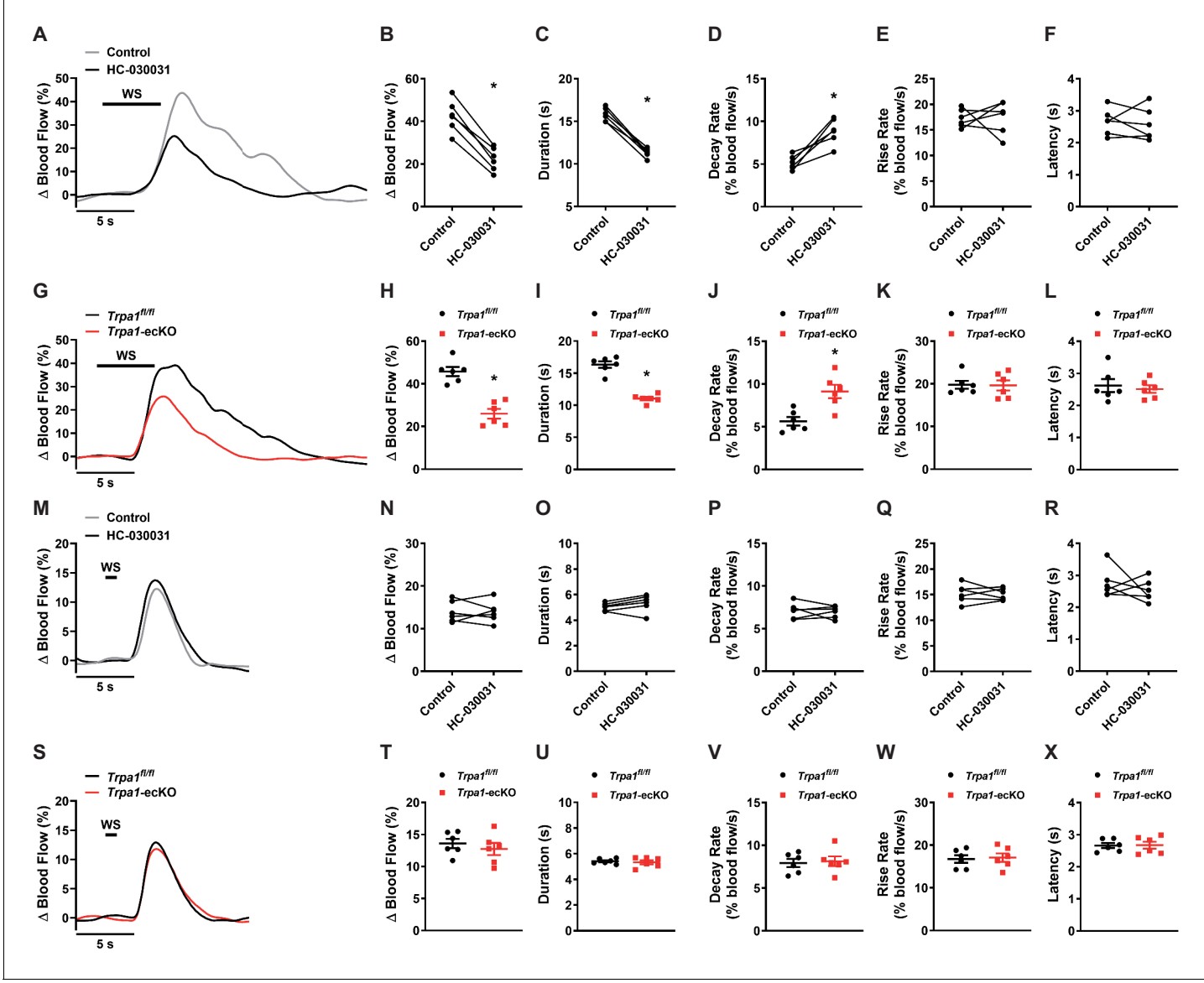

**Figure 8.** Functional hyperemia is dependent on brain capillary TRPA1 channels. (**A and B**) Representative traces (**A**) and summary data (**B**) showing the hyperemic response in the somatosensory cortex following contralateral whisker stimulation (WS) for 5 s, measured using laser-Doppler flowmetry in wild-type mice. Treatment with HC-030031 (100 mg/kg, *i.p.* for 30 min) reduced the hyperemic response (n = 6 animals; *p<0.05, paired *t*-test). (**C–F**) Duration (**C**), decay rate (**D**), rise rate (**E**), and latency (**F**) of the hyperemic response was also determined, and a reduction in duration and increase in decay rate was observed in animals treated with HC-030031 (100 mg/kg, *i.p.* for 30 min) (n = 6 animals; *p<0.05, paired *t*-test). (**G and H**) Representative traces (**G**) and summary data (**H**) showing the hyperemic response is blunted in *Trpa1*-ecKO mice compared with *Trpa1*^{fl/fl} controls following contralateral whisker stimulation for 5 s (n = 6 animals; *p<0.05, unpaired *t*-test). (**I to L**) Duration (**I**), decay rate (**J**), rise rate (**K**), and latency (**L**) of the hyperemic response was also determined, and duration was diminished and decay rate was increased in *Trpa1*-ecKO compared with *Trpa1*^{fl/fl} mice (n = 6 animals; *p<0.05, unpaired *t*-test). (**M and N**) Representative traces (**M**) and summary data (**N**) showing the hyperemic response was unchanged following a 1 s whisker stimulation in animals treated with HC-030031 (100 mg/kg, *i.p.* for 30 min) (n = 6 animals; paired *t*-test). (**O–R**) Duration (**O**), decay rate (**P**), rise rate (**Q**), and latency (**R**) were unaffected by HC-030031 treatment (100 mg/kg, *i.p.* for 30 min) (n = 6 animals; paired *t*-test). (**S and T**) Representative traces (**S**) and summary data (**T**) showing the hyperemic response did not differ between *Trpa1*^{fl/fl} and *Trpa1*-ecKO mice following a 1 s whisker stimulation (n = 6 animals; unpaired *t*-test). (**U–X**) Duration (**U**), decay rate (**V**), rise rate (**W**), and latency (**X**) also did not differ between *Trpa1*^{fl/fl} and *Trpa1*-ecKO mice (n = 6 animals; unpaired *t*-test).

The online version of this article includes the following source data and figure supplement(s) for figure 8:

**Source data 1.** Individual data points and analysis summaries for datasets shown in *Figure 8*.

**Figure supplement 1.** Experimental illustration of how functional hyperemia was measured in vivo.

**Figure supplement 2.** Lack of a hyperemic response following ipsilateral whisker stimulation.

*Figure 8 continued on next page*

*Figure 8 continued*

**Figure supplement 2—source data 1.** Individual data points and analysis summaries for datasets shown in *Figure 8—figure supplement 2*.

**Figure supplement 3.** Functional hyperemic response following 2 s whisker stimulation.

**Figure supplement 3—source data 1.** Individual data points and analysis summaries for datasets shown in *Figure 8—figure supplement 3*.

activation of Panx1 channels, thus establishing a propagating intercellular $Ca^{2+}$ signaling pathway (*Barbe et al., 2006*; *Suadicani et al., 2009*). Our data suggest that a similar signaling pathway operates in the cerebral capillary network, providing a novel short-range intercellular communication system between neighboring capillary endothelial cells. Our data further suggest that $Ca^{2+}$ influx through TRPA1 channels is the initiator of this pathway, but it is possible that other $Ca^{2+}$-permeable channels present on capillary endothelial cells can activate this pathway to initiate propagative vasodilation if they form a signaling complex with Panx1.

Purinergic signaling is an important endogenous pathway that regulates many cellular processes, including regulation of vascular tone in multiple vascular beds. Activation of ATP-gated, cation-conducting purinergic $P_2$ receptors and G protein-coupled receptors results in dual vasoconstriction and vasodilation responses. Sympathetic nerves surrounding vessels of numerous vascular beds contain ATP, which upon release, acts primarily on $P_2X_1$ receptors to promote $Ca^{2+}$ influx and subsequent contraction of vascular smooth muscle cells (*Burnstock, 2017*). Endothelial cells are a significant source of ATP that is released in response to certain stimuli, such as shear stress (*Schwiebert et al., 2002*). Secreted ATP activates endothelial $P_2X$ and $P_2Y$ receptors in an autocrine and/or paracrine manner to produce the endothelium-dependent vasodilators nitric oxide and prostacyclin as well as endothelium-dependent hyperpolarizing factors (*Burnstock, 2017*). Our data demonstrates that ATP derived from capillary endothelial cells activate $P_2X$ receptors on adjacent endothelial cells producing a propagating $Ca^{2+}$ signal that mediates dilation of upstream arterioles. However, there are other sources of ATP in the cerebral circulation. ATP released from surrounding astrocytes (*Kisler et al., 2017*; *Pascual et al., 2005*) and neurons (*Fields and Burnstock, 2006*; *Kisler et al., 2017*) can activate ATP-gated receptors on arteriolar vascular smooth muscle cells, resulting in contraction; it can also induce endothelium-dependent vasodilation through the release of paracrine factors (*Chen et al., 2014*; *Ralevic and Dunn, 2015*). Our data support a model in which endogenous ATP derived from astrocytes and/or neurons may induce dilation of upstream arterioles. Astrocytes and/or neurons that encompass capillaries may release ATP that directly activates $P_2X$ receptors expressed on capillary endothelial cells to initiate Panx1-dependent conducted vasodilation, thereby providing an additional capillary-based mechanism for initiation of NVC.

Our findings indicate that post-arteriole transitional segments are vital for functional hyperemic responses in the brain. These transitional vascular segments are composed of endothelial cells encompassed by specialized mural cells called ensheathing pericytes (*Grant et al., 2019*; *Hartmann et al., 2015*; *Hill et al., 2015*). Ensheathing pericytes are morphologically distinct from other types of mural cell types as they possess a protruding cell body and are elongated (*Grant et al., 2019*; *Hartmann et al., 2015*). Recent reports suggest that ensheathing pericytes are contractile and regulate the diameter of post-arteriole transitional segment to redistribute blood to neuronally active sites (*Hill et al., 2015*). Our findings suggest that post-arteriole transitional segments serve two additional functions during NVC. First, focal stimulation of these segments with substances that activate $K_{ir}$ or TRPA1 channels initiates conducted dilation of upstream arterioles. These data suggest that the brain's vascular sensory web extends throughout the capillary tree to include the post-arteriole transitional segment. Interestingly, the mechanism of ATP-induced signal propagation is unique to more distal reaches of the capillary network, as focal application of ATP onto the post-arteriole transitional region failed to dilate upstream arterioles. Secondly, our data show that post-arteriole transitional segments are critically important for the translation of slowly propagating short-range $Ca^{2+}$ signals originating deeper in the capillary bed into fast electrical signals that cause dilation of upstream parenchymal arterioles. Stimulation of IK and SK channels within the transitional region resulted in fast-propagating conducted dilation, indicating functional expression of these channels within this vascular segment. These functional data suggest that endothelial cells that occupy the post-arteriole transitional region are phenotypically distinguishable from those

found in the distal capillary tree, but a meticulous single-cell analysis is needed to verify this at the molecular level.

The NVC process is vital for maintaining cerebral blood flow to active neuronal sites. Notably, insufficient blood flow is a hallmark of many neurological disorders, including vascular cognitive impairment, stroke, and Alzheimer's disease. The demonstration that capillaries act as sensors in the control of microcirculatory hemodynamics has altered our view of functional hyperemia (*Longden et al., 2017*). Our findings extend this paradigm, demonstrating that TRPA1 channels and PPADS-sensitive purinergic receptors on brain capillary endothelial cells are capable of dilating upstream parenchymal arterioles to increase RBC flux and functional hyperemia in the brain. Our findings further show that signals initiated by TRPA1 channels and purinergic receptors propagate through capillary endothelial cells through a novel $Ca^{2+}$ signaling pathway that depends on endothelial Panx1 channels. These $Ca^{2+}$ signals are converted to fast-conducting electrical signals by SK and IK channels in the post-arteriole transition region. We conclude that multiple sensory mechanisms present in brain endothelial cells are vital for the regulation of cerebral blood flow and functional hyperemia and could be targeted to treat cerebrovascular dysfunction.

# Materials and methods

**Key resources table**

| Reagent type (species) or resource | Designation | Source or reference | Identifiers | Additional information |
|---|---|---|---|---|
| Genetic reagent (*M. musculus*) | C57BL/6J | Jackson Laboratory | Stock #: 000664 RRID:IMSR_JAX:000664 | |
| Genetic reagent (*M. musculus*) | *Panx1*-ecKO | Dr. Brant Isakson PMID:26242575 | | |
| Genetic reagent (*M. musculus*) | *Tek^Cre* | Jackson Laboratory | Stock #: 008863 RRID:IMSR_JAX:008863 | |
| Genetic reagent (*M. musculus*) | *Trpa1^fl/fl* | Jackson Laboratory | Stock #: 008654 RRID:IMSR_JAX:008654 | |
| Genetic reagent (*M. musculus*) | *Cdh5*-GCaMP8 | CHROMus (https://chromus.vet.cornell.edu/cdh5gcamp8/) PMID:23240011 | | |
| Antibody | Cy3 conjugated anti-α-smooth muscle actin (Mouse monoclonal) | Sigma-Aldrich, Inc | Cat. #: C6198 RRID:AB_476856 | (1:200, 1.0–1.5 mg/ml) |
| Other | Alexa Fluor 488 conjugated isolectin B4 | ThermoFisher Scientific | Cat. #: I21411 | (1:200, 1.0 mg/ml) |
| Other | Alexa Fluor 568 conjugated isolectin B4 | ThermoFisher Scientific | Cat. #: I21412 | (1:200, 1.0 mg/ml) |
| Other | Alexa Fluor 633 conjugated hydrazide | ThermoFisher Scientific | Cat. #: A30634 | (1:1000, 1.0 mg/ml) |
| Other | DAPI Fluoroshied mounting medium | Abcam plc. | Cat. #: ab104139 | |
| Other | Fluorescein isothiocyanate (FITC)-dextran | Sigma-Aldrich, Inc | Cat. #: FD150S | |
| Other | Tetramethylrhodamine isothiocyanate (TRITC)-dextran | Sigma-Aldrich, Inc | Cat. #: T1287 | |
| Peptide, recombinant protein | Apamin | Tocris Bioscience | Cat. #: 1652 | |
| Peptide, recombinant protein | Apyrase | Sigma-Aldrich, Inc | Cat. #: A6535 | |
| Peptide, recombinant protein | Collagenase type I | Worthington Biochemical Corporation | Cat. #: LS004194 | |
| Peptide, recombinant protein | Elastase | Worthington Biochemical Corporation | Cat. #: LS002292 | |

*Continued on next page*

*Continued*

| Reagent type (species) or resource | Designation | Source or reference | Identifiers | Additional information |
|---|---|---|---|---|
| Peptide, recombinant protein | Neutral protease | Worthington Biochemical Corporation | Cat. #: LS02104 | |
| Chemical compound, drug | 4-hydroxynonenal (4-HNE) | Cayman Chemical | Cat. #: 32100 | |
| Chemical compound, drug | Adenosine 5-triphosphate (ATP) disodium salt | Sigma-Aldrich, Inc | Cat. #: A2383 | |
| Chemical compound, drug | Allyl isothiocyanate (AITC) | Sigma-Aldrich, Inc | Cat. #: 377430 | |
| Chemical compound, drug | HC-030031 | Tocris Bioscience | Cat. #: 2896 | |
| Chemical compound, drug | Indomethacin | Sigma-Aldrich, Inc | Cat. #: I7378 | |
| Chemical compound, drug | NS309 | Tocris Bioscience | Cat. #: 3895 | |
| Chemical compound, drug | N$\omega$-Nitro-L-arginine methyl ester (L-NAME) hydrochloride | Sigma-Aldrich, Inc | Cat. #: N5751 | |
| Chemical compound, drug | PPADS tetrasodium salt | Tocris Bioscience | Cat. #: 0625 | |
| Chemical compound, drug | Tamoxifen | Sigma-Aldrich, Inc | Cat. #: T5648 | |
| Software, algorithm | pClamp software | Molecular Devices, LLC. (http://www.moleculardevices.com/products/software/pclamp.html) | RRID:SCR_011323 | Version 10.2 |
| Software, algorithm | FluoView FV1000 FV10-ASW software | Olympus (https://www.olympus-lifescience.com/en/support/downloads/) | RRID:SCR_014215 | Version 4.02 |
| Software, algorithm | GraphPad Prism software | GraphPad Software, Inc (https://www.graphpad.com/) | RRID:SCR_002798 | Version 8.2 |
| Software, algorithm | ImageJ software | National Institutes of Health (https://imagej.nih.gov/ij/) | RRID:SCR_003070 | Version 1.52 n |
| Software, algorithm | IonWizard software | IonOptix, LLC. (https://www.ionoptix.com/products/software/ionwizard-core-and-analysis/) | | Version 6.4.1.73 |
| Software, algorithm | VisiView software | Visitron Systems GmbH (https://www.visitron.de/products/visiviewr-software.html) | | Version 4.5.0.7 |
| Software, algorithm | µManager software | University of California, San Francisco (https://micro-manager.org/) | RRID:SCR_000415 | Version 1.4.22 |

## Animals

Adult (12–16 weeks of age) male and female mice were used for all experiments. All animal procedures used in this study were approved by the Institutional Animal Care and Use Committee of the University of Nevada, Reno, School of Medicine (protocol number: 20-06-1020). C57BL/6J mice (Jackson Labs, stock number: 000664) were used as wild-type controls in this study. For all *Cre* lines used in this study, only male mice expressing *Cre*-recombinase were used as breeders when generating cell-specific transgenic mice. Endothelial cell-specific deletion of TRPA1 was achieved by initially crossing mice homozygous for *Trpa1* containing loxP sites flanking S5/S6 transmembrane domains (Jackson Labs, stock number: 008654) with heterozygous *Tek^Cre* mice (Jackson Labs, stock number: 008863) to produce intermediate heterozygote mice which were used to generate *Trpa1*-ecKO mice, as previously described (*Sullivan et al., 2015*). Mice homozygous for floxed *Trpa1*, but without expression of *Cre*-recombinase (*Trpa1^fl/fl*), were used as controls for experiments involving *Trpa1*-ecKO mice. Mice with tamoxifen-inducible endothelial cell-specific knockout of Panx1

channels (*Panx1*-ecKO), provided by Dr. Brant Isakson (University of Virginia, USA), were generated as previously described (*Lohman et al., 2015*). *Cre*-recombinase was induced by intraperitoneal (*i. p.*) injection of animals with tamoxifen (1 mg in 0.1 ml of peanut oil; Sigma-Aldrich, Inc, USA) on 10 consecutive days. A recombinase efficacy greater than 95% has been reported following tamoxifen treatment for this *Cre* line (*Monvoisin et al., 2006*). Mice expressing *Cre* alone treated with tamoxifen and *Panx1*-ecKO mice treated with vehicle were used as controls. *Cdh5*-GCaMP8 mice, expressing the genetically encoded $Ca^{2+}$ indicator GCaMP8 exclusively in the endothelium, were developed by CHROMus (Cornell University, USA) (*Ohkura et al., 2012*). When appropriate, mice were randomly allocated into groups using a randomizer, and genotypes were masked to the investigator during group allocation and data collection. Mice were euthanized by decapitation under isoflurane anesthesia. The brain was isolated into a solution of ice-cold, $Ca^{2+}$-free, $Mg^{2+}$-based physiological saline solution (Mg-PSS) containing 5 mM KCl, 140 mM NaCl, 2 mM $MgCl_2$, 10 mM HEPES, and 10 mM glucose (pH 7.4, NaOH; all salts from Sigma-Aldrich, Inc).

## Isolation of native capillary endothelial cells

Individual capillary endothelial cells were isolated as previously described (*Longden et al., 2017*). Brains were denuded of surface vessels with an aCSF-wetted cotton swab, and two 1-mm-thick brain slices were excised and homogenized in ice-cold aCSF using a Dounce homogenizer. The brain homogenate was filtered through a 70 µM filter, and capillary networks that were captured on the filter were transferred to a new tube. Individual capillary endothelial cells were isolated by enzymatic digestion with 0.5 mg/ml neutral protease (Worthington Biochemical Corporation, USA) and 0.5 mg/ml elastase (Worthington Biochemical Corporation) in endothelial cell isolation solution composed of 5.6 mM KCl, 55 mM NaCl, 80 mM sodium glutamate, 2 mM $MgCl_2$, 0.1 mM $CaCl_2$, 4 mM glucose, and 10 mM HEPES (pH 7.3; all salts from Sigma-Aldrich, Inc) for 45 min at 37°C. After the first digestion, 0.5 mg/ml collagenase type I (Worthington Biochemical Corporation) was added, and a second 2-min incubation at 37°C was performed. Digested networks were washed in ice-cold endothelial cell isolation solution, then triturated with a fire-polished glass Pasteur pipette to produce individual endothelial cells.

## Whole-cell patch-clamp electrophysiology

Whole-cell patch-clamp electrophysiology was used to assess the presence of functional TRPA1 channels on native brain capillary endothelial cells. Following isolation, capillary endothelial cells were transferred to a recording chamber and allowed to adhere to glass coverslips for 10 min at room temperature (~22°C).

TRPA1 currents were recorded in cells patch-clamped in the conventional whole-cell configuration, and currents were amplified using an Axopatch 200B amplifier (Molecular Devices, LLC., USA). Currents were filtered at 1 kHz and digitized at 10 kHz. Pipettes were fabricated from borosilicate glass (1.5 mm outer diameter, 1.17 mm inner diameter; Sutter Instruments, USA), fire-polished to yield a tip resistance of 3–6 MΩ and filled with a solution consisting of 10 mM NaOH, 11.4 mM KOH, 128.6 mM KCl, 1.091 mM $MgCl_2$, 3.206 mM $CaCl_2$, 5 mM EGTA, 10 mM HEPES, and 5 mM ruthenium red (pH 7.2; all salts from Sigma-Aldrich, Inc). The bath solution consisted of 134 mM NaCl, 6 mM KCl, 2 mM $MgCl_2$, 10 mM HEPES, 4 mM glucose, and 1 mM EGTA (pH 7.4; all salts from Sigma-Aldrich, Inc). Currents were induced by a ramp protocol (−100 to +100 mV over 300 ms). The mean capacitance of capillary endothelial cells was 6.83 ± 2.2 pF (n = 7 cells from three animals). TRPA1 currents were recorded following the addition of 4-HNE (100 nM; Cayman Chemical, USA) to the external bath solution. The selective TRPA1 antagonist HC-030031 (10 µM; Tocris Bioscience, USA) was used to validate the current.

IK, SK, and $K_{ir}$ currents were recorded using the conventional whole-cell configuration at a holding potential of −50 mV, with 400 ms ramps from −100 to +45 mV. The external bathing solution was composed of 134 mM NaCl, 6 mM KCl, 1 mM $MgCl_2$, 2 mM $CaCl_2$, 10 mM glucose, and 10 mM HEPES. The composition of the pipette solution was 10 mM NaCl, 30 mM KCl, 10 mM HEPES, 110 mM $K^+$ aspartate and 1 mM $MgCl_2$ (pH 7.2). IK and SK currents were activated by adding NS309 (10 µM; Tocris Bioscience) to the external bath solution. $K_{ir}$ channel currents were recorded with elevated extracellular $[K^+]$ in the external bath solution (60 mM KCl, 80 mM NaCl). $BaCl_2$ (10 µM; Sigma-Aldrich, Inc) was used to isolate $K_{ir}$ currents.

Clampex and Clampfit software (pClamp version 10.2; Molecular Devices, LLC.) were used for data acquisition and analysis, respectively. All recordings were performed at room temperature.

## Pressure myography

Pressure myography was performed using the recently described arteriole-capillary microvascular preparation (*Longden et al., 2017*), consisting of a capillary segment with attached intact capillaries (*Figure 2A*). Briefly, a 3 mm x 5 mm x 3 mm (W x L x D) section of the brain surrounding the middle cerebral artery was dissected and placed into a microdissection dish containing ice-cold Mg-PSS. The middle cerebral artery and the surrounding pial meninge were carefully removed so as to maintain branching parenchymal arterioles within the brain tissue. Thereafter, parenchymal arterioles with attached capillaries were carefully blunt-dissected from the underlying cerebral tissue and transferred to a pressure myograph chamber (Living Systems Instrumentation, USA) containing oxygenated (21% $O_2$, 6% $CO_2$, 73% $N_2$; Praxair Technology, Inc, USA) aCSF (124 mM NaCl, 3 mM KCl, 2 mM $CaCl_2$, 2 mM $MgCl_2$, 1.25 mM $NaH_2PO_4$, 26 mM $NaHCO_3$, 4 mM glucose; all salts from Sigma-Aldrich, Inc) and a Sylgard pad at the chamber base. Microvascular preparations were mounted between two glass micropipettes (outer diameter ~20–40 µm) and secured with nylon monofilaments. The outermost tips of attached capillaries were pinned onto the surface of the Sylgard pad to immobilize the capillary bed and allow pressurization of the preparation. Intraluminal pressure was controlled using a servo-controlled peristaltic pump (Living Systems Instrumentation). Arteries were visualized with an inverted microscope (Accu-Scope Inc, USA) coupled to a USB camera (The Imaging Source LLC., USA). Intraluminal diameter as a function of time was recorded using IonWizard software (version 6.4.1.73; IonOptix, LLC., USA). Microvascular preparations were bathed in warmed (37°C), oxygenated aCSF at an intraluminal pressure of 5 mmHg. Following a 15-min equilibration period, intraluminal pressure was increased to 40 mmHg and spontaneous tone was allowed to develop. Preparations that developed less than 10% tone were discarded. Localized application of drugs onto the capillary bed was achieved by placing a micropipette attached to a Picospritzer III (Parker Hannifin Corporation, USA) adjacent to capillary extremities. The viability of attached capillaries was assessed by locally applying a 7 s pulse of aCSF containing elevated [$K^+$] (10 mM) onto capillary extremities. Preparations that failed to dilate to elevated [$K^+$] were discarded.

In control experiments designed to validate the microvascular preparation, the connection between the upstream parenchymal arteriole segment and attached capillaries was severed. In other control experiments, spatial spread was determined by pulsing aCSF containing 1% w/v Evans Blue dye (Sigma-Aldrich, Inc). The role of TRPA1 channels was examined by locally applying 4-HNE (1 µM) or AITC (30 µM; Sigma-Aldrich, Inc) to the attached capillaries, and the role of purinergic signaling was examined by focal application of ATP (10 µM; Sigma-Aldrich, Inc). NS309 (10 µM) was also applied to determine expression of IK and SK channels. In separate experiments, preparations were modified by removing attached capillary segments and leaving the post-arteriole transitional segment intact. In these experiments, the post-arteriole transitional segment was directly stimulated via a micropipette placed adjacent to this segment. Underlying mechanisms were examined by adding pharmacological agents to the superfusing bath solution. Apyrase (1 U/mL), $BaCl_2$ (30 µM), indomethacin (10 µM), and L-NAME (100 µM) were purchased from Sigma-Aldrich, Inc, and apamin (300 nM), HC-030031 (10 µM), PPADS (10 µM), and TRAM34 (1 µM) were purchased from Tocris Bioscience. Lumen diameter was continuously recorded, and responses were expressed as vasodilation (%). Velocity data were obtained by determining the latency of the response following local application of compounds in relation to the distance traveled between the capillary extremity or post-arteriole transitional segment to the parenchymal arteriole.

## Fluorescence imaging

Images of the ex vivo microvascular preparation were obtained from wild-type mice. Following isolation and cannulation, preparations were fixed (4% formaldehyde (Sigma-Aldrich, Inc) in phosphate buffered saline (PBS), 20 min at room temperature) and blocked (0.5% Triton X-100 (Sigma-Aldrich, Inc), 5% SEABlock Blocking Buffer (ThermoFisher Scientific, USA) in PBS, 2 hr at room temperature). Preparations were labeled with Alexa Fluor 488 conjugated isolectin B4 (1:200 of a 1 mg/ml stock; catalogue number: I21411, ThermoFisher Scientific), Cy3 conjugated anti-α-smooth muscle actin antibody (1:200 of a 1–1.5 mg/ml stock; catalogue number: C6198, Sigma-Aldrich, Inc), and Alexa

Fluor 633 conjugated hydrazide (1:1000 of a 1 mg/ml stock; catalogue number: A30634, Thermo-Fisher Scientific) for 2 hr at room temperature. Images were obtained using a custom-built upright microscope (Olympus BX51 WI; Olympus Corp., Japan) equipped with an 89 North LDI laser system (LDI-NIR 15030, 89 North, Inc, USA), and an ORCA-Fusion Digital CMOS C15440-20UP camera (Hamamatsu Corporation, Japan). Images of the complete microvascular preparation were obtained using a 10x water-immersion objective (numerical aperture 0.3, Olympus), and magnified images of the different vascular segments using a 40x water-immersion objective (numerical aperture 0.8, Olympus). Each field of view was 1371 × 1523 pixels (10x, one pixel = 0.65 μm; 40x, one pixel = 0.16 μm). Z-stacks were captured at 1 μm intervals to allow for three-dimensional reconstruction of the microvascular preparation. Images were captured using VisiView software (version 4.5.0.7, Visitron Systems GmbH, Germany) and analyzed using ImageJ software (version 1.52 n, National Institutes of Health, USA).

The length of capillary endothelial cells was determined by fluorescence labeling isolated capillary networks. Isolated capillary networks were placed onto a glass coverslip and fixed (4% formaldehyde in PBS, 20 min at room temperature) and blocked (0.5% Triton X-100, 5% SEABlock Blocking Buffer in PBS, 2 hr at room temperature). Capillary networks were visualized by staining preparations with Alexa Fluor 568 conjugated isolectin B4 (1:200 of a 1 mg/ml stock (catalogue number: I21412, ThermoFisher Scientific), 2 hr at room temperature). Aqueous mounting medium with DAPI (catalogue number: ab104139, Abcam plc., USA) was used to visualize nuclei of endothelial cells. Fluorescence images were taken by an Olympus FluoView laser scanning confocal microscope (Olympus FluoView FV1000, Olympus) equipped with a Showa Optronics laser system (GLS5414A, Showa Optronics Co., Ltd, Japan) and a 60x oil-immersion objective (numerical aperture 1.42, Olympus). Each field of view was 1024 × 1024 pixels (one pixel = 0.21 μm). Z-stacks were captured at 0.5 μm intervals to allow for three-dimensional reconstruction of the capillary networks. Images were captured using FluoView FV1000 FV10-ASW software (version 4.02, Olympus), and analyzed using ImageJ software. The length of capillary endothelial cells was determined by measuring the distance between nuclei.

## $Ca^{2+}$ imaging

Microvascular preparations isolated from *Cdh5*-GCaMP8 mice were mounted in a pressure myograph chamber, as described above. Changes in capillary endothelial cell $[Ca^{2+}]$ were recorded in real time after local exposure to TRPA1 agonists or ATP. Videos were recorded at an average of 23 frames/s for approximately 120 s (2760 frames). Baseline $Ca^{2+}$ activity was recorded for the first 60 s (~1380 frames), then the capillary bed was picospritzed with AITC (30 μM) or ATP (10 μM), and the preparations were recorded for an additional 60 s (~1380 frames). Videos were obtained using a custom-built upright microscope (Olympus BX51 WI) equipped with epifluorescence illumination (CoolLED pE-300^white, CoolLED Ltd., UK), a 20x water-immersion objective (numerical aperture 0.5, Olympus), and an ORCA-Fusion Digital CMOS C14440-20UP camera. Each field of view was 1152 × 1152 pixels (one pixel = 0.65 μm). Videos were captured using μManager software (version 1.4.22, University of California, San Francisco, USA) and analyzed using ImageJ software. Discrete regions of interest (ROIs) along the capillary network were selected for analysis. The fractional increase in fluorescence ($F/F_0$) was determined for ROIs, where fluorescence (F) is normalized to basal fluorescence ($F_0$). Underlying mechanisms were examined by adding pharmacological agents to the superfusing bath solution. Velocity data were obtained by determining the latency of the response between adjacent ROIs following local application of compounds in relation to the distance traveled.

### Two-photon imaging of in vivo brain microcirculatory hemodynamics

Hemodynamics of the murine microcirculation were assessed in vivo after capillary endothelial cell TRPA1 activation as described previously (*Longden et al., 2017*). Briefly, mice were anesthetized with isoflurane (5% induction, 2% maintenance) and the skull was exposed. Thereafter, a stainless-steel head plate was attached over the right hemisphere using dental adhesive, and the head was immobilized by securing the head plate to a holding frame. A cranial window (~2 mm diameter) was made in the skull above the somatosensory cortex, after which FITC-labeled dextran (150 kDa; 150 μL of a 3 mg/mL solution; Sigma-Aldrich, Inc) was intravenously (*i.v.*) administered to allow visualization of the cortical cerebral vasculature and contrast imaging of RBCs. Following completion of the surgical procedure, isoflurane anesthesia was replaced with combined α-chloralose (50 mg/kg, *i.p.*;

Sigma-Aldrich, Inc) and urethane (750 mg/kg, *i.p.*; Sigma-Aldrich, Inc) to eliminate confounding vaso-dilatory effects of isoflurane. Capillaries downstream of arterioles were identified and selected for study. A micropipette was positioned adjacent to the capillary, such that AITC (30 µM) was directly applied onto the capillary. Tetramethylrhodamine isothiocyanate (TRITC; 150 kDa; 0.2 mg/mL; Sigma-Aldrich, Inc)-labeled dextran was also included in the micropipette to visualize the micropi-pette location and determine the spatial coverage of the solution. The duration and pressure of fluid ejection via a glass micropipette was calibrated to obtain a localized application area approximately 10 µm in diameter. Images were obtained using a Zeiss LSM-7 two-photon microscope (Zeiss, USA), coupled to a Coherent Chameleon Vision II Titanium-Sapphire pulsed infrared laser (Coherent, USA) and a 20x Plan Apochromat water-immersion objective (numerical aperture 1.0, Zeiss). Each field of view was 512 × 512 pixels (one pixel = 0.42 µm). After excitation at 820 nm, emitted FITC-labeled dextran and TRITC-labeled dextran fluorescence was separated through 500–550 and 570–610 nm bandpass filters, respectively. RBC flux was determined by line-scan imaging of the capillary. Change in RBC flux through the stimulated capillary and cross-sectional area of upstream arterioles were obtained using ImageJ software.

## Functional hyperemia

After anesthetizing mice with isoflurane (5% induction, 2% maintenance), the skull was exposed and the head was immobilized in a stereotaxic frame. The skull of the right hemisphere was carefully thinned using a drill to visualize the surface vasculature of the somatosensory cortex. Following com-pletion of the surgical procedure, isoflurane anesthesia was replaced with combined $\alpha$-chloralose (50 mg/kg, *i.p.*) and urethane (750 mg/kg, *i.p.*) to eliminate confounding vasodilatory effects of isoflur-ane. Perfusion was monitored via a laser-Doppler flowmetry probe (PeriFlux System PF5000, Perimed AB, Sweden) positioned above the somatosensory cortex. The contralateral whiskers were stimulated for either 1, 2, or 5 s, and changes in perfusion were recorded. Contralateral whiskers were stimulated three times at 2 min intervals. Ipsilateral whiskers were also stimulated as a control for potential vibration artifacts. The role of TRPA1 channels was assessed by treating mice with HC-030031 (100 mg/kg, *i.p.* for 30 min), and recording the hyperemic response in *Trpa1*-ecKO mice. Data are presented as changes in perfusion relative to baseline, calculated as follows: %$\Delta$ Blood flow = (perfusion during stimulus/baseline perfusion)×100.

## Statistical analysis

All data are expressed as means ± standard error of the mean (SEM), unless specified otherwise. The value of 'n' refers to number of cells for patch-clamp electrophysiology and fluorescence imaging experiments, the number of vessel preparations for pressure myography and fluorescence and $Ca^{2+}$ imaging experiments, and the number of animals for two-photon imaging studies and functional hyperemia assessments. Experimental sample size was determined using a two-sided power analysis to reach a power of 0.8 for an $\alpha$ of 0.05. We estimated a minimum of five cells per group for patch-clamp electrophysiology experiments, five vessel preparations per group for pressure myography and $Ca^{2+}$ imaging experiments, and four animals per group for studies involving two-photon imag-ing and functional hyperemia.

Statistical analyses and graphical presentations were performed using GraphPad Prism software (version 8.2, GraphPad Software, Inc, USA). Statistical analyses were performed using Students paired or unpaired two-tailed *t*-test or one-way analysis of variance (ANOVA) with Tukey's multiple comparisons test. A value of $p < 0.05$ was considered statistically significant.

## Acknowledgements

The authors thank Dr. Albert L Gonzales for the use of his custom-built upright microscope in acquir-ing the fluorescent images of the microvascular preparation.

The present study was supported by grants from the National Heart, Lung, and Blood Institute (R35HL155008, R01HL091905, R01HL137852, R01HL139585 and R01HL146054 to SE; K99HL140106 to PWP; P01HL120840 and R01HL137112 to BEI), the National Institute of Neurological Disorders and Stroke (RF1NS110044 and R61NS115132 to SE), the National Institute of General Medical Scien-ces (P20GM130459 to SE), and the American Heart Association Postdoctoral Fellowship award (20POST35210155 to AM). The Transgenic Genotyping and Phenotyping Core at the COBRE Center

for Molecular and Cellular Signaling in the Cardiovascular System, University of Nevada, Reno is maintained by a grant from NIH/NIGMS (P20GM130459 Sub#5451). The High Spatial and Temporal Resolution Imaging Core at the COBRE Center for Molecular and Cellular Signaling in the Cardiovascular System, University of Nevada, Reno, is maintained by a grant from NIH/NIGMS (P20GM130459 Sub#5452).

## Additional information

### Funding

| Funder | Grant reference number | Author |
|---|---|---|
| National Heart, Lung, and Blood Institute | R01HL091905 | Scott Earley |
| National Heart, Lung, and Blood Institute | R01HL137852 | Scott Earley |
| National Heart, Lung, and Blood Institute | R01HL139585 | Scott Earley |
| National Heart, Lung, and Blood Institute | R01HL146054 | Scott Earley |
| National Heart, Lung, and Blood Institute | K99HL140106 | Paulo W Pires |
| National Heart, Lung, and Blood Institute | P01HL120840 | Brant E Isakson |
| National Heart, Lung, and Blood Institute | R01HL137112 | Brant E Isakson |
| National Institute of Neurological Disorders and Stroke | RF1NS110044 | Scott Earley |
| National Institute of Neurological Disorders and Stroke | R61NS115132 | Scott Earley |
| National Institute of General Medical Sciences | P20GM130459 | Scott Earley |
| National Heart, Lung, and Blood Institute | R35HL155008 | Scott Earley |
| American Heart Association | 20POST35210155 | Amreen Mughal |

The funders had no role in study design, data collection and interpretation, or the decision to submit the work for publication.

### Author contributions

Pratish Thakore, Conceptualization, Data curation, Formal analysis, Validation, Investigation, Visualization, Methodology, Writing - original draft, Project administration, Writing - review and editing; Michael G Alvarado, Data curation, Formal analysis, Investigation, Visualization; Sher Ali, Amreen Mughal, Paulo W Pires, Evan Yamasaki, Harry AT Pritchard, Data curation, Formal analysis, Investigation; Brant E Isakson, Cam Ha T Tran, Resources, Methodology; Scott Earley, Conceptualization, Resources, Data curation, Supervision, Funding acquisition, Validation, Investigation, Visualization, Methodology, Project administration, Writing - review and editing

### Author ORCIDs

Pratish Thakore (iD) https://orcid.org/0000-0002-2086-5453
Michael G Alvarado (iD) http://orcid.org/0000-0002-3489-9021
Amreen Mughal (iD) http://orcid.org/0000-0002-0046-2286
Paulo W Pires (iD) http://orcid.org/0000-0001-5972-4554
Scott Earley (iD) https://orcid.org/0000-0001-9560-2941

## Ethics

Animal experimentation: All animal procedures used in this study were approved by the Institutional Animal Care and Use Committee of the University of Nevada, Reno, School of Medicine (protocol number: 20-06-1020).

## Decision letter and Author response

Decision letter https://doi.org/10.7554/eLife.63040.sa1
Author response https://doi.org/10.7554/eLife.63040.sa2

# Additional files

## Supplementary files

• Transparent reporting form

## Data availability

All data generated or analyzed during this study are included in the manuscript and supporting files. Source data files have been provided for all figures and figure supplements.

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
