## [Decision Letter]

**Acceptance summary:**

Vasodilation in response to an increase in neuronal activity in the brain is governed by multiple mechanisms. This study beautifully describes a specific mechanism of capillary dilation mediated by TRPA1 channels, a mechanism that may be in particular important in brain disease.

**Decision letter after peer review:**

Thank you for submitting your article "Brain Endothelial Cell TRPA1 Channels Initiate Neurovascular Coupling" for consideration by *eLife*. Your article has been reviewed Kenton Swartz as the Senior and Reviewing Editor, and three reviewers. The following individuals involved in review of your submission have agreed to reveal their identity: Andy Shih (Reviewer #2); Anna Devor (Reviewer #3).

The reviewers have discussed the reviews with one another and the Reviewing Editor has drafted this decision to help you prepare a revised submission.

Summary:

The editors and reviewers all agreed that this is an impressive, mechanistic study of capillary-to-arteriole signal conduction during neurovascular coupling. Building on seminal earlier work from Longden et al., this study now demonstrates that TRPA1-Panx1-P2X signaling controls a distinct Ca^2+^-dependent signal propagation at the level of capillaries. This signal is slower than K^+^ conduction, but similarly moves upstream until it reaches the post-arteriole transition segment. At this point, TRPA1 signaling passes the baton to faster K^+^ signaling that leads to rapid and widespread dilation of upstream arterioles. The study is very comprehensive and well controlled, spanning many techniques, such as ex vivo isolated vessel preparations for detailed mechanistic testing to in vivo two-photon imaging and laser doppler for physiological relevance. The mechanism is teased out with endothelial specific TRPA1 deletions and a battery of pharmacological studies. Considering the myriad signaling mechanisms that may be involved in NV coupling, it makes sense that there are multiple convergent mechanisms for capillary-to-arteriole conduction. That these mechanisms may function on distinct temporal scales is very interesting and opens the door to deeper studies. Overall, we are very enthusiastic about the work and think a suitably revised manuscript would be appropriate for publication in *eLife*.

Essential revisions:

1) The careful attention to distinct microvascular zones is appreciated. It would be helpful to better link the capillary-to-arteriole preparation with in vivo vascular architecture. Mural cell morphology is currently used, which is great, but the cell shapes are a hard to make out in the images. Without information on branching order from the penetrating arteriole or α-SMA expression, it is hard to relate to what is known in vivo. There is also some concern that the transitional segment might include some capillary zone, as it is quite long as depicted (~500 μm in Figure 6A). in vivo, this zone is typically shorter (~50-150um) before it branches into capillaries, at a point where α-SMA content in the mural cell drops substantially. What would help is α-SMA immunostaining, if at all possible for this preparation, using a directly conjugated antibody from Σ (Σ: C6198). We also suggest using Alexa 633 (elastin staining) to show where the post-arteriole transitional segment begins.

Related to this, a depiction of where precisely the parenchymal arteriole and its capillary extremities is extracted from the brain slice would be helpful. Also, if possible, some discussion on the variation between each ex vivo preparation. For example, at times, the transitional segment in cortical vascular networks can be very short, or much longer depending upon its diameter at the point of offshoot from the penetrating arteriole.

2) Figure 7B. The latency to onset of RBC flux increase is delayed by what seems like 5 seconds. This is interesting and consistent with a slow propagation through the capillary bed. The reporting of the metrics could therefore go deeper. For instance, what is the mean latency? What is the rate of flux increase once it begins? This may be beyond the scope, but it would be nice to have a K^+^ induced comparison for the in vivo two-photon imaging studies, so we can understand the difference between slow and fast mechanisms. It would also serve as a good positive control. This is because the speed of vasodilation can be dependent on anesthesia or state of physiology, which can differ between labs.

3) Figure 7G,H. It may be more apt to measure cross-sectional area of the penetrating arteriole rather than diameter. It will result in a more robust change than diameter. Diameter is also tricky if the cross-section is an ellipse. You may wish to also better explain what the images are showing with a schematic, for readers not familiar with two-photon imaging.

4) A deeper discussion of why a slow propagation mechanism is needed would be helpful. NV coupling is typically considered to be a process on the timescale of seconds. Perhaps, the slow calcium response helps to sustain or buffer the rapid dilatory response from electrical conduction? Can this be gathered from your laser doppler experiments? In short, what part of the hemodynamic response function is affected with loss of TRPA1 signaling? The model suggests TRPA1 should affect slow components, but the doppler data shows a dampened response across the entire stimulation period and in fact affects the initial dilation. To see this hemodynamic response better, we think you should use shorter stimuli (i.e. 1 second) because 20 seconds of continuous whisker stimulation is too long and muddies the response function.

5) Related to point 4, while its clear that dilation is governed by multiple mechanisms, not all of these mechanisms are necessarily active at once, and some may be reserved for pathological conditions (which does not make them less important to understand). It is not clear to us whether the described pathway is relevant to normal functional hyperemia or dilation. The data show ~ 20 seconds delay in the in vivo drug application experiment (Figure 7). This implies that the TRPA1 pathway would not play a role for the first 10-20 seconds following an onset on neuronal activity in response to a stimulus. However, the following in vivo experiment where whiskers are stimulated for 30 seconds (Figure 8) shows a reduction in the CBF response throughout the stimulus. This suggests to us that the reduction in the CBF response in the KO is not directly due to the mechanism described in previous figures. In the discussion, the authors talk about studies of TRPA1 channels under pathological conditions published by their own team and others. Given the inconsistency between the 20-seconds delay in Figure 7 and reduction if CBF response from the start of the response in the KO in Figure 8, the authors might consider doing a hypoxia experiment instead, where TRPA1 may play an obvious role. The animals would need to be ventilated to avoid hyperventilation as a way to counteract hypoxia (this happens despite anesthesia). Another possibility would be to remove the last experiment (in vivo CBF) that only adds confusion due to discrepancies in the time course (the delay issue). You should emphasize in the Discussion that further studies are needed to determine whether the TRPA1 mechanism is active during normal functional hyperemia or vasodilation under pathophysiological conditions.

[Editors' note: further revisions were suggested prior to acceptance, as described below.]

Thank you for resubmitting your work entitled "Brain Endothelial Cell TRPA1 Channels Initiate Neurovascular Coupling" for further consideration by *eLife*. Your revised article has been evaluated by Kenton Swartz as the Senior and Reviewing Editor.

Summary:

The reviewers agree that you have done an outstanding job of revising the manuscript; the quality and novelty of the work is very high. The new histology provided in Figure 3A-D is excellent. The in vivo blood flow data showing lack of attentuation with shorter stimulation is consistent with your conclusions. One reviewer has two suggestions for final revisions that we hope will be straightforward to address:

1) There is still potential for confusion in understanding vascular zones in Figure 3. The schematic in Figure 3G looks anatomically like Figure 3A-D. However, it is really depicting just the distal end of what is shown in the histology. What confused me was that the horizontal vessel in Figure 3A-D is a parenchymal arteriole (acta2 expressing, elastin containing), not the post-arteriole transitional segment, which is the offshoot that projects vertically above the arteriole at the distal end. So, the inset image for parenchymal arteriole could be placed on the horizontal vessel. Or the histology images could be turned 90 degrees clockwise to be more consistent with the schematic. Also clarify in Figures 2A and Figure 6A, are the horizontally oriented vessels post-arteriole transitional segments?

2) Subsection “Functional hyperemia in the somatosensory cortex requires endothelial cell TRPA1 channels”. "dependent" is misspelled. Also, reference the figure that shows the 7 seconds delay in this sentence.

---

## [Author Response]

Essential revisions:1) The careful attention to distinct microvascular zones is appreciated. It would be helpful to better link the capillary-to-arteriole preparation with in vivo vascular architecture. Mural cell morphology is currently used, which is great, but the cell shapes are a hard to make out in the images. Without information on branching order from the penetrating arteriole or α-SMA expression, it is hard to relate to what is known in vivo. There is also some concern that the transitional segment might include some capillary zone, as it is quite long as depicted (~500 μm in Figure 6A). in vivo, this zone is typically shorter (~50-150um) before it branches into capillaries, at a point where α-SMA content in the mural cell drops substantially. What would help is α-SMA immunostaining, if at all possible for this preparation, using a directly conjugated antibody from Σ (Σ: C6198). We also suggest using Alexa 633 (elastin staining) to show where the post-arteriole transitional segment begins.Related to this, a depiction of where precisely the parenchymal arteriole and its capillary extremities is extracted from the brain slice would be helpful. Also, if possible, some discussion on the variation between each ex vivo preparation. For example, at times, the transitional segment in cortical vascular networks can be very short, or much longer depending upon its diameter at the point of offshoot from the penetrating arteriole.

To clarify, only the distal end of the branch in Figure 6A is the transitional segment.

The proximal end that is closest to the cannulated arteriole is a smaller branch arteriole and contains circumferential smooth muscle. We have replaced the images in Figure 3 with images obtained from preparations labeled with the reviewer recommended conjugated anti-αsmooth muscle actin antibody and the elastin stain Alexa Fluor^TM^ 633 hydrazide. In these new images, we observe α-smooth muscle actin and elastin labeling in the arteriole segment and the proximal end of the branch stemming from it. Anti-α-smooth muscle actin also labeled ensheathing pericytes within the transitional segment as observed by the clear bump-on-a-log morphology. Very little to no α-smooth muscle actin labeling was observed in the capillary region. Hydrazide staining was also absent in the transitional segment and capillary region, further differentiating these regions from the arteriole segment. We have included a depiction of where the preparations are derived from (Figure 2A) and have provided information regarding the variability between each preparation, including information about the transitional segment and number and branch order of capillaries.

2) Figure 7B. The latency to onset of RBC flux increase is delayed by what seems like 5 seconds. This is interesting and consistent with a slow propagation through the capillary bed. The reporting of the metrics could therefore go deeper. For instance, what is the mean latency? What is the rate of flux increase once it begins? This may be beyond the scope, but it would be nice to have a K^+^ induced comparison for the in vivo two-photon imaging studies, so we can understand the difference between slow and fast mechanisms. It would also serve as a good positive control. This is because the speed of vasodilation can be dependent on anesthesia or state of physiology, which can differ between labs.

The mean latency was 7.5 ± 1.2 seconds, and the rate of flux was 11.6 ± 2.3 RBC/seconds. We have included these values in the revised manuscript. We understand the reviewer's rationale for having a direct comparison to the K^+^ induced response. The method used to generate our data was the same as those used by Longden et al., and we, therefore, have decided to make comparisons to their published data, for example the reported latency was 3.8 ± 0.9 seconds.

3) Figure 7G,H. It may be more apt to measure cross-sectional area of the penetrating arteriole rather than diameter. It will result in a more robust change than diameter. Diameter is also tricky if the cross-section is an ellipse. You may wish to also better explain what the images are showing with a schematic, for readers not familiar with two-photon imaging.

We agree and have modified Figure 7G to I to show changes in cross-sectional area. We have also included an illustration of what the images are showing as part of Figure 7A.

4) A deeper discussion of why a slow propagation mechanism is needed would be helpful. NV coupling is typically considered to be a process on the timescale of seconds. Perhaps, the slow calcium response helps to sustain or buffer the rapid dilatory response from electrical conduction? Can this be gathered from your laser doppler experiments? In short, what part of the hemodynamic response function is affected with loss of TRPA1 signaling? The model suggests TRPA1 should affect slow components, but the doppler data shows a dampened response across the entire stimulation period and in fact affects the initial dilation. To see this hemodynamic response better, we think you should use shorter stimuli (i.e. 1 second) because 20 seconds of continuous whisker stimulation is too long and muddies the response function.

We agree and performed a series of laser Doppler experiments using stimuli of 1, 2, and 5 seconds and analyzed the kinetics of response. We have reported the latency, duration, rise rate, and decay rate in the revised manuscript. Following a 5-second stimulation, we found that HC-030031 treated animals and Trpa1-ecKO mice had blunted hemodynamic responses and increased the decay rate compared with controls. Interestingly no differences in hemodynamic responses between these groups were observed when contralateral whiskers were stimulated for or 1 or 2 seconds. We have modified Figure 8 to show these new findings. These new data suggest that the TRPA1 pathway is only recruited under more prolonged periods of neuronal activity and contributes to sustaining the blood flow increase. We believe the K_ir_ pathway described by Longden et al. is responsible for rapidly increasing blood flow during these shorter stimulations and that the slower TRPA1 pathway is not evoked during brief stimulation.

5) Related to point 4, while its clear that dilation is governed by multiple mechanisms, not all of these mechanisms are necessarily active at once, and some may be reserved for pathological conditions (which does not make them less important to understand). It is not clear to us whether the described pathway is relevant to normal functional hyperemia or dilation. The data show ~ 20 seconds delay in the in vivo drug application experiment (Figure 7). This implies that the TRPA1 pathway would not play a role for the first 10-20 seconds following an onset on neuronal activity in response to a stimulus. However, the following in vivo experiment where whiskers are stimulated for 30 seconds (Figure 8) shows a reduction in the CBF response throughout the stimulus. This suggests to us that the reduction in the CBF response in the KO is not directly due to the mechanism described in previous figures. In the discussion, the authors talk about studies of TRPA1 channels under pathological conditions published by their own team and others. Given the inconsistency between the 20-seconds delay in Figure 7 and reduction if CBF response from the start of the response in the KO in Figure 8, the authors might consider doing a hypoxia experiment instead, where TRPA1 may play an obvious role. The animals would need to be ventilated to avoid hyperventilation as a way to counteract hypoxia (this happens despite anesthesia). Another possibility would be to remove the last experiment (in vivo CBF) that only adds confusion due to discrepancies in the time course (the delay issue). You should emphasize in the Discussion that further studies are needed to determine whether the TRPA1 mechanism is active during normal functional hyperemia or vasodilation under pathophysiological conditions.

We apologize for the incorrect latency value reported in the original version of the manuscript. The correct value is 7.5 ± 1.2 seconds. Furthermore, our new functional hyperemia experiments indicate that the TRPA1 pathway is vital for maintaining the increase in blood flow, which supports the mechanism identified. Studying the effect of TRPA1-induced hyperemic response under hypoxic conditions is a good idea and would make an excellent topic for a follow-up paper.

[Editors' note: further revisions were suggested prior to acceptance, as described below.]

1) There is still potential for confusion in understanding vascular zones in Figure 3. The schematic in Figure 3G looks anatomically like Figure 3A-D. However, it is really depicting just the distal end of what is shown in the histology. What confused me was that the horizontal vessel in Figure 3A-D is a parenchymal arteriole (acta2 expressing, elastin containing), not the post-arteriole transitional segment, which is the offshoot that projects vertically above the arteriole at the distal end. So, the inset image for parenchymal arteriole could be placed on the horizontal vessel. Or the histology images could be turned 90 degrees clockwise to be more consistent with the schematic. Also clarify in Figure 2A and Figure 6A, are the horizontally oriented vessels post-arteriole transitional segments?

We have rotated the images 90 degrees clockwise so that it is more consistent with the schematic shown in Figure 3G. We have also included a new set of images of the offshoot branch arteriole which we hope will alleviate potential confusion. In both Figure 2A and Figure 6A, the proximal end that is closest to the cannulated arteriole is an offshoot branch arteriole. The transitional segment is located at the distal end, similar to what is observed in our fluorescent images. We have clarified this in the figures and the respective legends in the revised manuscript.

2) Subsection “Functional hyperemia in the somatosensory cortex requires endothelial cell TRPA1 channels”. "dependent" is misspelled. Also, reference the figure that shows the 7 seconds delay in this sentence.

Thank you – we have corrected this typo. Originally this data was mentioned in text, but we have now demonstrated this data as a supplemental figure (Figure 7—figure supplement 1) in the revised manuscript.